

# Comparison of diurnal aerosol products retrieved from combinations of micro-pulse lidar and sun-photometer over KAUST observation site

Anton Lopatin[1], Oleg Dubovik[2], Georgiy Stenchikov[3], Ellsworth Judd Welton[4], Illia Shevchenko[3], David Fuertes[1], Marcos Herreras-Giralda[1], Tatsiana Lapyonok[2] and Alexander Smirnov[4]

[1]GRASP SAS, Lezennes, 59260, France
[2]Laboratoire d'Optique Atmospherique, UMR 8518, Villeneuve d'Ascq, 59650, France
[3]King Abdullah University of Science and Technology, Physical Science and Engineering Division, Thuwal, 23955-6900, Saudi Arabia
[4]NASA Goddard Flight Center, Greenbelt, MD 20771, USA

*Correspondence to*: Anton Lopatin (anton.lopatin@grasp-sas.com)

**Abstract.** This study focuses on comparison of aerosol columnar AOD and Lidar Ratios together with vertical profiles of aerosol extinction and backscatter at 532 nm retrieved over the King Abdullah University of Science and Technology (KAUST) campus observation site for a period of 2019–2022 using GRASP and MPLNET approaches. An emphasis is made on independent analysis of daylight and nighttime retrievals to estimate how strongly the differences in assumptions of both methods made in absence of nighttime AOD observations influence the retrieval results. Additionally, two aerosol products provided by GRASP excluding and including the volume depolarization observations at 532 nm provided by MPLNET are analyzed to estimate the potential benefits of usage of depolarization data in aerosol profile retrievals.

In overall, both columnar and vertical MPLNET and GRASP products demonstrated a better agreement for day-time retrievals for the GRASP product excluding the depolarization information. At the same time, inclusion of the volume depolarization observations improved the agreement between MPLNET and GRASP estimated values at nighttime, both columnar and vertical.

In addition, estimated values of daytime extinction profiles at a ground level were compared to assess the impact of assumptions of constant aerosol vertical distribution in the cut-off zone of lidar observations implied in GRASP. The values estimated by GRASP demonstrated a good agreement with MPLNET, both for retrievals including and excluding volume depolarization information.

A seasonal variability of diurnal cycle of aerosol properties estimated by GRASP over KAUST site for the period 2019–2022 is presented, analyzed and discussed.



## 1 Introduction

Vertical distribution of atmospheric aerosols plays an important role in effects that define aerosol influence on the Earth's climate. This includes both direct aerosol-radiation interaction that affects earth radiative budget and indirect effects through the modification of cloud formation and their lifecycle (Twomey,1974; Albrecht et al., 1989) as well as via a semi-direct effect could be mentioned that consists of modifying the cloud formation by the change of atmospheric temperature due to the direct

absorption of solar light by aerosols (Koch and Del Genio, 2010).

In addition, the exposure to aerosol particles is also known to impact human health (Ault and Axson, 2017). High concentrations of fine particulates in air increases health risks associated with respiratory (Pope et al., 2002) and cardiopulmonary functions (Wellenius et al., 2012) as well as lung tumors (Raaschou-Nielsen et al., 2013). In these regards knowledge about aerosol distribution near the surface becomes crucial to estimate long-term exposure to particulates and

therefore the health risks for the population. Additionally, the knowledge about aerosol vertical distribution is crucial for verification and tuning of the chemical transport and climate calculation models both on regional and global scales.

Remote sensing offers the most suitable observations for this purpose. In fact, remote sensing techniques are capable to characterize the properties of ambient, non-perturbed aerosols and can provide continuous data at regional and even global scales. However, it's important to note that different types of remote sensing measurements have varying sensitivities and often

yield complementary information about aerosols, which require meticulous analysis and appropriate interpretation in order to maximize the benefits of the observations. LiDAR (Light Detection and Ranging) is one of the most common remote sensing techniques that allows to observe aerosol vertical variation as well as it's temporal evolution. Lidar detectors have the capability to measure the time delay between emission of the light pulse, usually provided by a laser, and it's return that is backscattered from the aerosol particles. This allows one to establish the location and by measuring the magnitude of the returned signal the

particle concentration of the aerosol layer. Measured lidar data from backscatter lidars are inversely proportional to the range squared, and depends on the emitted laser energy and other lidar specific calibration factors (overlap, laser-detector crosstalk or afterpulse, and polarization quality) as well as the solar background at the laser wavelength. Lidar processing methods must calibrate and normalize the measured data to produce the so-called lidar signal (here referred to as normalized relative backscatter, NRB) at the specified wavelength, described by the following equation:

$$L_{NRB}(\lambda, h) = C\beta(\lambda, h) exp\left(-2\int_{h_{min}}^{h} \sigma(\lambda, h')dh'\right), \tag{1}$$

Where $\sigma(\lambda, h) = \sigma_a(\lambda, h) + \sigma_m(\lambda, h)$ is extinction, $\beta(\lambda, h) = \beta_a(\lambda, h) + \beta_m(\lambda, h)$ is backscatter of aerosol layer, that both contain molecular and aerosol parts correspondingly and $C$ is a so-called calibration constant that is a function of the receiver efficiency, aperture, and optical design. The other standard form of the lidar signal removes dependence on $C$, through either laboratory calibration or normalization against molecular background, and is referred to as attenuated backscatter. However,

determination of aerosol backscatter and extinction profiles using typical backscatter lidar retrievals (Fernald et al., 1972; Klett, 1981; Fernald, 1984) are independent of $C$, and thus either form of the lidar signal may be used. GRASP utilizes a



different approach, with normalization of the NRB signal to exclude the influence of the calibration constant $C$ (Lopatin et al., 2013, 2021):

$$L(\lambda, h) = \frac{L_{NRB}(\lambda, h)}{\int_{h_{min}}^{h_{max}} L_{NRB}(\lambda, h\prime) dh\prime} \qquad (2)$$

The lidar equation (Eq. 1) depends on two, strictly speaking, independent profiles of extinction and backscatter rendering retrieval of these properties from the single observation impossible. There are a variety of methods that allow for overcoming such limitation by introducing in various forms additional, usually a priori, information about relation between aerosol extinction and backscatter, making the solution of lidar equation possible. In the simplest approach it takes the form of a linear dependence parameter, known as lidar ratio (LR):


$$S(\lambda) = \frac{\sigma(\lambda)}{\beta(\lambda)} = \frac{4\pi}{\omega_o(\lambda) P_{11}(180°, \lambda)}, \qquad (3)$$

where $\omega_o$ is the single scattering albedo, $P_{11}(180°)$ is the phase function at 180-degree backscatter angle, providing the lidar ratio in units of steradian ($Sr$).

One of the most straightforward estimations was proposed by Klett (Klett, 1981) and consists in assuming the lidar ratio to be vertically constant and fixed to a selected value, which is usually equal to 50 $Sr$ or is chosen following the properties of the

expected aerosol type. E.g., Cloud–Aerosol Lidar with Orthogonal Polarization (CALIOP) aerosol typing algorithm (Kim et al., 2018) could be considered as the further advancement of Klett approach, allowing to assign several pre-defined lidar ratios to different aerosol types based on climatological values. However, large errors in the retrieved aerosol backscatter and extinction profiles can occur if the assigned lidar ratio differs from the actual value. Another technique is utilized to reduce these errors, whereby independent measurements of the aerosol optical depth (AOD) are used to constrain a backward Fernald

(1984) retrieval of the aerosol profiles (Welton et al., 2000), ensuring the retrieved extinction profile integrates to the measured AOD. With this technique, a column averaged lidar ratio is calculated from the measurements, instead of being pre-assigned. Errors can still occur since the lidar ratio is assumed constant through the atmospheric layer analyzed, but the results are expected be more accurate due to the AOD constraint.

Another option is improvement of observation techniques in order to perform measurements of extinction and backscatter

separately by so called Raman techniques (Wandinger et al., 2005). Such systems, together with the backscatter signal, can directly measure the attenuation of the atmosphere by triggering radiation emission by certain gases at different atmospheric layers which provide direct sensitivity to the amount of aerosol below the level of the induced emission. As a further development of such techniques High Spectral Resolution Lidars (HSRL; e.g., Hair et al., 2008) should be additionally mentioned. HSRL allows measurement of aerosol attenuation and backscatter separately at a much closer wavelengths than

traditional Raman techniques, which usually rely on assumptions of aerosol Angstrom exponent in order to process correctly Raman shifted signals in a combination with elastic channels.

Such sophisticated lidar systems as Raman and HSRL greatly enhance the information available from lidar observations of aerosol properties. However, even the most advanced lidars have limitations when it comes to capturing fine details of aerosol





characteristics compared, for example, to passive multi-angular observations. This is partly because lidar systems typically
utilize just a few spectral channels (between 1 and 5) and can register intensity and state of depolarization of reflected signals
with the number of independent measurements summing up to no more than eight, even for the most advanced setups.
Furthermore, ground-based lidar observations have a blind zone close to the ground due to afterpulse and incomplete
geometrical overlap between the laser beam and telescope field of view, which can extend from several hundred meters to
several kilometers depending on the system's design and purpose. Additionally, the signals captured by lidar are typically weak
and dim significantly with distance, therefore lidar measurements are subject to significant registration noise, particularly
during daytime observations, which can limit the capabilities of Raman or HSRL observations in daylight. As such, it is always
desirable to have ancillary data from collocated photometric measurements to aid the interpretation of lidar observations, and
to recognize the complementary nature of passive and active measurements, even with the use of advanced lidar systems.

Various algorithms have been proposed for joint processing of coincident photometric and lidar ground-based observations to
retrieve aerosol properties. Some of these methods focus on treating the data of available lidar systems combined into networks.
In this study we utilize observations and aerosol data provided by the Micro Pulse Lidar network (MPLNET; Welton et al.,
2001; Welton et al., 2018). MPLNET began operations in 2000 with the goal of providing collocated lidar profiling at key
sites in the NASA Aerosol Robotic Network (AERONET) (Holben et al., 1993). MPLNET aerosol processing utilizes the
constrained retrieval technique (Welton et al., 2000) with the AERONET AOD as the constraint, providing vertical
distributions of aerosol optical properties, notably extinction and backscatter and calculation of a column average lidar ratio.
The retrieval techniques and limitations from using a single wavelength backscatter lidar preclude retrieval of microphysical
parameters such as size and refractive index. However, MPLNET lidars have been polarized (Flynn et al., 2007; Welton et al.,
2018) since 2014-2015, and thereafter the aerosol processing includes retrievals of the aerosol depolarization ratio, providing
additionally information on particle shape.

Other algorithmic techniques attempt to advance and derive vertical profiles of several aerosol components, as well as extra
parameters of the column-integrated properties of aerosols. For example, the Lidar and Radiometer Inversion Code (LiRIC;
Chaikovsky et al., 2016) and Generalized Aerosol Retrieval from Radiometer and Lidar Combination/Generalized Retrieval
of Atmosphere and Surface Properties (GARRLiC/GRASP; Lopatin et al., 2013, 2021) algorithms use joint data from a multi-
wavelength lidars and AERONET Sun–sky-scanning radiometers. The LiRIC, for example, use microphysical columnar
properties provided by AERONET as necessary a priori values in order to perform retrieval of aerosol vertical profiles.
However, in such approach the columnar properties retrieval is not benefiting from any extra sensitivity of lidar measurements,
and rely on several additional assumptions, e.g., spectral interpolation of complex refractive index (Chaikovsky et al., 2016).

Thus, this article focuses on comparison of aerosol columnar and vertical optical products retrieved over the King Abdullah
University of Science and Technology (KAUST) campus observation site using different methodologies, specifically using
GRASP and MPLNET approaches. Both approaches have several similarities. For example, they use the same set of lidar
signals from MPLNET standard processing, and produce aerosol profiles of extinction and backscatter at 532 nm together with
estimations of columnar AOD and lidar ratio (LR) at the same wavelength. Also, both methods provide estimations performed



during day and night time. Unfortunately, KAUST AERONET site doesn't provide lunar AOD observations, and therefore such retrieval scheme will be out of scope of this study. Indeed, the use of nighttime lunar observations could significantly
improve the nocturnal retrievals requiring less assumptions to be made regarding the aerosol temporal variability in case of GRASP. Lack of AERONET lunar AOD also has a significant impact on the nighttime MPLNET aerosol processing, as described below. Instead, the study will focus on comparing the columnar and vertical values of nighttime retrievals in order to estimate how well the assumptions of different methodologies agree with each other and how strongly existing differences influence the retrieval results.

## 2 Dataset description and methodology

The KAUST campus is situated in Thuwal on the eastern coast of the Red Sea, in the western Arabian Peninsula (22.3∘ N, 39.1∘ E). The region experiences local dust storms that arise from the surrounding inland deserts (e.g., see, Kalenderski and Stenchikov, 2016), as well as distant dust from northeastern Africa through the Tokar Gap (Parajuli et al., 2020). Consequently, there is a year-round presence of desert dust in the atmosphere over the site.

Since 2014, a Micro-Pulse Lidar has been in operation at KAUST site. This lidar is co-located with the eponymous AERONET site and is part of the Micro-Pulse Lidar Network (Welton et al., 2001, 2018). MPLNET Version 3 (V3) data products are automatically processed, providing near real time (NRT) data generated with NRT calibrations. MPLNET utilizes the same product level convention as AERONET. Level 1 and 1.5 data are NRT but with the latter including quality assurance screening. Final Level 2 products are generated after Level 2 AERONET data are available, and using final calibrations. Meteorological
data from the NASA GEOS-5 model are used to calculate molecular quantities and diagnostic parameters. The MPLNET product suite includes the signal product (NRB) which are the lidar signals, volume depolarization ratio, and diagnostics (Campbell et al., 2002; Welton and Campbell, 2002; Welton et al., 2018). The MPLNET cloud product (CLD) includes multiple cloud layer heights and tops, cloud phase, and estimates of thin cloud optical depth (Lewis et al., 2016, 2020). The MPLNET aerosol product (AER) includes aerosol layer height; profiles of extinction, backscatter, aerosol depolarization ratio;
the columnar lidar ratio; and calculation of the lidar constant ($C$) (Welton et al., 2000, 2002, 2007, 2018). The aerosol variables are retrieved continuously using a running 20-minute, cloud-screened signal average (where cloud screening is only applied to clouds below the aerosol top height), and re-gridded to a 1-minute temporal grid in the product. The MPLNET boundary layer product (PBL) contains mixed layer heights and estimates of the mixed layer AOD (Lewis et al., 2013). All data products are open to public access on the MPLNET website (https://mplnet.gsfc.nasa.gov), and are stored in netcdf4, CF compliant
formats. All variables in each product contain uncertainties derived from error propagation of raw data and calibrations. More detailed information about MPLNET data is available on the website. Here, comparisons are made between MPLNET V3 standard aerosol product retrievals (L1.5 AER) and those produced from GRASP using corresponding MPLNET L1.5 NRB signal data as described below.



KAUST is unique lidar site on the Red Sea coast, and its co-location with the AERONET station allows for a more accurate
retrieval of the vertical profile of aerosols (Welton et al., 2000; Parajuli et al., 2020; Lopatin et al., 2021). Additionally, KAUST
has a meteorological station that performs measurements of air temperature, humidity, wind speed, and incoming short-wave
and long-wave radiative fluxes. Stations that measure various parameters of interest for dust-related research, such as dust
deposition rate, vertical profile, near-surface concentration, and spectral optical depth, are particularly rare across the global
dust belt. The collection of these co-located data provides unique opportunity to obtain a more comprehensive understanding
of dust emissions and transport in the region.

Almost three consecutive years of data starting from march 2019 till December 2022 collected over KAUST observation site
including vertical profiles of volume depolarization provided by MPLNET lidar in combination with co-located AERONET
observations were processed using the recent operational version (1.1.1) of GRASP software. All available AERONET and
MPLNET V3 Level 1.5 NRB signal data were processed at once taking into account non-simultaneity of data acquisition by
lidar and sun-photometer. More specifically, a so-called multi-pixel (Dubovik et al., 2011, 2021; Lopatin et al., 2021) approach
was used. In this methodology additional a priori limitations on time variability of aerosol parameters are applied.

Additionally, a recently established concept of aerosol modeling allowing to infer some information about aerosol composition
from remote sensing observation (Li et al., 2019) was used. Specifically, the so-called GRASP/Components approach, where
the spectrally dependent complex refractive indices for both fine and coarse modes of aerosol are modeled using an internal
mixture of different chemical components with known spectral dependencies of the complex refractive index, was used. These
components include black and brown carbon, fine and coarse insoluble dust material, coarse absorbing insoluble components,
mainly represented by iron oxides that are commonly present in the desert dust and determines its absorbing properties, fine
and coarse non-absorbing solubles that represent anthropogenic and natural salts, notably sulfur and ammonia, as well as
aerosol water content. A Maxwell-Garnett effective medium approximation (e.g., Schuster et al., 2016a,b, 2009, 2005)
and direct volume mixture can be used to estimate the effective refractive index, by combining insoluble components into a
host media that contains soluble components diluted in water (e.g., see Fig. 1).

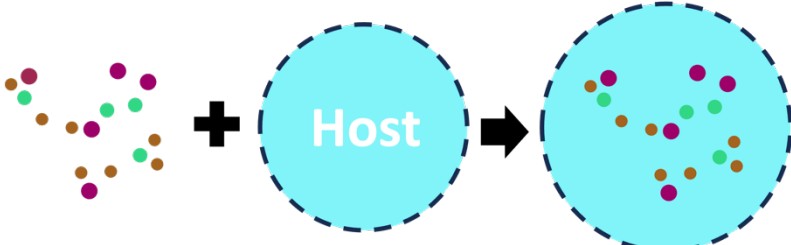

**Figure 1: The illustration of the general concept of modeling of effective refractive index using the Maxwell Garnett effective medium**
**approximation (adapted from Li et al., 2019).**



The approach has been applied to AERONET data which has allowed deriving aerosol optical properties, such as AOD (Aerosol Optical Depth), AE (Angström Exponent), size distribution and SSA (Single Scattering Albedo) that are fully consistent with the conventional AERONET retrieval (Li et al., 2019, Zhang, et al., 2022). At the same time, this approach
provides some insight about the aerosol type (e.g., desert dust, biomass burning, urban polluted or clean aerosol, sea salt, etc.). Moreover, the approach allows for identifying some variability within each type of aerosol (e.g., level of absorption, spectral dependence, etc.). It should be noted, despite that Maxwell-Garnett approximation was exploited in the above application, the linear volume mixing approach is also realized in GRASP and demonstrates valuable features (Li et al., 2019).

In addition, the present study has also considered the possibility of using a priori knowledge on temporal continuity of aerosol
properties evolution to be used as additional constraints in similar ways as described by Lopatin et al., (2021). Specifically, a priori knowledge about the temporal continuity of aerosol properties evolution was used as an additional constraint on temporal variability of aerosol chemical composition, sphericity fraction and size distribution. All available AERONET measurements collected on the day before (starting at noon) and day after (before noon) were used in combination with available day and night MPLNET lidar measurements of normalized relative backscatter signal and volume depolarization. This allows achieving
improvements in the daytime AERONET and lidar retrievals for the observations close to noon, when sun-photometer observations provide rather limited constraints for the retrieval due to observation geometry with reduced coverage of scattering angles (that are usually more pronounced at low latitude sites such as KAUST), resulting in a higher amount of data observations processed. At night only lidar measurements were conducted. The measurements taken at 17:00, 20:00, 23:00 and 02:00 UTC were used.

For GRASP processing, lidar signals were copped and treated within 270 — 5670 m altitude range, containing 73 vertical strobes at the MPLNET 75 m vertical resolution (https://mplnet.gsfc.nasa.gov/product-info/product_pages.cgi?p=NRB). An accumulation of 15 minutes prior and after the mentioned times (including the times of evening and morning AERONET observations) was applied to the MPLNET profiles. If AERONET observations (both AOD and almucantar) were not available due to cloud contamination or any other reason, the NRB profiles around this
observation times were also discarded. The newest available MPLNET V3 aerosol data of Level 1.5 were used for the comparisons. The details on data screening and quality assurance provided by MPLNET team could be found in (https://mplnet.gsfc.nasa.gov/versions.htm) and (https://mplnet.gsfc.nasa.gov/product-info/). Since nighttime observations can't rely on AERONET cloud screening as daytime observations, a cloud screening based on V3 L1.0 CLD/cloud_base product was used for nighttime observations, thus effectively discarding the profiles that contained cloud
base values within the altitude crop area. Observations containing unphysical values of volume depolarization (negative of higher than 100%) within the altitudes of interest were also discarded. The combined AERONET MPLNET data were treated following two scenarios: excluding and including volume depolarization profiles at 532 nm, referenced below as scenario 1 and scenario 2 respectively. Accounting for volume depolarization (Welton et al., 2018) allows GRASP to use an extended aerosol microphysical model that distinguishes between some of the properties of fine and coarse aerosol particles (see Table
2 for details).



Table 1 summarizes instruments configurations of measurement times used for combined MPLNET AERONET retrievals using GRASP. The details of MPLNET data preparation and combined retrievals could be found in (Lopatin et al., 2021). It should be outlined that values provided in the column "Estimated measurement uncertainty" do not represent exactly the uncertainty of the real observations, but are used in order to weight the observations in GRASP retrieval to properly account

for the uncertainty differences and information content of various types of observations. E.g., sky radiances provided by AERONET observations are crucial for estimating aerosol microphysical properties that support the inversion on lidar profiles, while the AOD measurements give good constraints on aerosol quantity which are both crucial to correct inversion of the lidar equation (Lopatin et al., 2013, 2021), therefore AERONET provided observations have lower estimated uncertainty in order to guarantee the convergence of the combined lidar-photometric data. All four observation uncertainties could be both

estimated on relative or absolute scale, with observations having a substantial dynamic range (SKY radiance and NRB) being estimated in relative scale, while AOD and volume depolarization (in percentage) are in absolute scale. At the same time, lidar profile uncertainty increases with range due to the lower signal-to-noise ratio, and therefore can't be estimated with a single value, with these regards, values provided in Table 1 for NRB and volume depolarization signals represent an altitude average residual that is desired to be achieved (or surpassed) during the inversion process. It should be noted that the GRASP weighted

uncertainty values used for MPLNET lidars are significantly higher than the actual measurement uncertainties derived from signal calibrations and measurement conditions for both the signal and volume depolarization. The actual measurement uncertainties for the data are provided by the MPLNET standard aerosol processing.

**Table 1: Summary of the data and their combinations used by the GRASP multi-temporal retrieval scheme.**

| Instrument | Measurement type | Estimated measurement uncertainty | Wavelength (nm) | Observation set diurnal period | |
|---|---|---|---|---|---|
| | | | | Daytime | Nighttime |
| Sun photometer | Atmosphere optical thickness | 0.01 (abs.) | 440, 670, 870, 1020 | YES | NO |
| | Almucantar | 5% (rel.) | 440, 670, 870, 1020 | YES | NO |
| MPL | Normalized relative backscatter profile | 30% (rel.) | 532 | Scenario 1 & Scenario2 | |
| | Volume depolarization ratio | 0.015 (abs.) | 532 | Scenario2 | |

Overall, 6450 for scenario 1 and 4380 for scenario 2 profiles were estimated from successfully processed combined MPLNET and sun-photometer data, covering the period of 23 March 2019 — 31 December 2022. The difference in the number of observations is explained by additional screening applied to the volume depolarization profiles, when the whole measurement combination was omitted for GRASP processing if no full volume depolarization profile within the altitude crop range was available after the accumulation within the ±15-minute window.




**Table 2: List of aerosol properties retrieved during GRASP AERONET+MPLNET inversion, parameters marked in bold are selected for the comparison.**

| Aerosol characteristic | Scenario 1 | Scenario 2 |
|---|---|---|
| Volume averaged size distribution (particle radii 0.05 – 15 μm) | Total | Fine & Coarse* |
| Volume fractions of aerosol chemical composition | Fine & Coarse | Fine & Coarse |
| Volume averaged complex refractive index at 440, 532, 670, 860 and 1020 nm | Fine & Coarse | Fine & Coarse |
| **Optical thicknessat** 440**, 532,** 670, 860 and 1020 **nm** | **Total,** Fine & Coarse | **Total,** Fine & Coarse |
| Absorption optical thickness at 440, 532, 670, 860 and 1020 nm | Fine & Coarse | Fine & Coarse |
| Volume averaged SSA at 440, 532, 670, 860 and 1020 nm | Fine & Coarse | Fine & Coarse |
| **Volume averaged Lidar Ratio at** 440**, 532,** 670, 860 and 1020 **nm** | **Total,** Fine & Coarse | **Total,** Fine & Coarse |
| Vertical profiles of aerosol mixing ratio, altitudes from 11 m (ground level) till 7670m** | Total | Fine & Coarse |
| **Vertical profiles of aerosol extinction at** 440**, 532,** 670, 860 and 1020 **nm**** | **Total** | **Total,** Fine & Coarse |
| Vertical profiles of aerosol absorption at 440, 532, 670, 860 and 1020 nm** | Total | Total |
| **Vertical profiles of aerosol backscatter at** 440**, 532,** 670, 860 and 1020 **nm**** | **Total** | **Total,** Fine & Coarse |
| Vertical profiles of aerosol SSA at 440, 532, 670, 860 and 1020 nm** | — | Total |
| Vertical profiles of aerosol lidar ratio at 440, 532, 670, 860 and 1020 nm** | — | Total |

*\* maximum radius of fine mode is 0.57 μm, minimal radius of coarse mode is 0.33 μm*

*\*\* the value of any aerosol property between ground level and minimal reliable altitude of lidar measurements (270 m a.s.l.) is considered to be constant*

Table 2 summarizes the estimations of aerosol properties provided by GRASP synergetic retrievals from total atmospheric optical depth and Sky radiance measurements in almucantar geometry from the sun-photometer at 4 (440, 670, 870 and 1020 nm) wavelengths in combination with normalized attenuated range-corrected backscatter (NRB) and volume depolarization ratio at 532 nm using the approach described above.

While, GRASP and MPLNET methodologies have several similar features, as it was mentioned above, they have a significant number of differences as well. For example, several following major differences in the data treatment between GRASP and MPLNET should be outlined. First of all, GRASP performs inversion of both datasets simultaneously, allowing the lidar signal to influence the photometric observations and vice versa, while MPLNET retrieval relies on AOD to constraint the solution of Fernald equation, which has to be interpolated into the operating wavelength of the lidar (532 nm), rendering the cross-influence of two observation types impossible. Secondly, GRASP despite initially emerging from AERONET retrieval has a



number of changes accumulated over years of development, including radiative transfer optimizations and inclusion of chemical components retrieval option, that replaces the direct retrieval of complex refractive index values, like in a "classic"

AERONET retrieval approach, by retrieval of fraction of the components with known spectral dependencies of refractive index, as well as a possibility to realize multi-pixel approach, etc.

Nonetheless, the usage of exactly the same lidar signal datasets as an input for both approaches and provision of directly comparable products in the form of vertical distributions of aerosol extinction and backscatter, as well as of vertically averaged columnar optical thickness and lidar ratio at 532 nm opens a unique opportunity for inter-comparison of these intrinsically

different methodologies. Also, both algorithms rely on temporal limitation of key aerosol properties in order to be able to treat nighttime data, when no sun-photometric observations were available for this site. In addition, the MPLNET aerosol data are provided continuously on a 1-minute grid as described above, utilizing an alternative method to estimate AOD and LR between available AERONET observations. Indeed, AOD in a standard AERONET configuration is provided approximately every 15 minutes, while the almucantar inversions which provide estimations of microphysical properties (size distribution, complex

refractive index and sphericity fraction) that are required to estimate columnar lidar ratio are performed only 8 to 10 times a day. It should be noted that MPLNET retrieval allows for using lunar AOD during nighttime, however these observations are not always available due to the changes in the lunar phase (Barretto et al., 2016) or due to the instrumentation limitations. As mentioned above, AERONET station at KAUST site wasn't equipped with the robotic photometer capable of performing lunar AOD observations therefore a standard procedure (https://mplnet.gsfc.nasa.gov/product-info/product_pages.cgi?p=AER ) was

used to estimate diurnal variations of AOD and LR with one minute time resolution.

The MPLNET aerosol retrievals are generated using two methods specified by the nature of the AOD. Observation times with available collocated AERONET AOD (daytime or lunar) utilize the retrieval approach described above. The lidar calibration constant, $C$, is also calculated during the retrieval using the independently measured AOD (Welton et al., 2002). This produces a discreet number of C values per day from available daytime and lunar observations. A continuous 1 minute-gridded C

variable is constructed by linearly interpolating between each discreet C value. For observations between AERONET measurements, the C value and the aerosol top height are used to calculate an effective column AOD from the lidar data and the molecular background. This requires a 1 km cloud free layer 500 meters above the top of the aerosol (the calibration zone). The AOD is then used as input to the same retrieval algorithm used for the collocated AERONET retrievals. This process produces three types of aerosol data in the MPLNET product: retrievals constrained by daytime AERONET AOD, lunar

AERONET AOD, or interpolated AOD. These are combined together in the 1-minute re-gridded data variables, with quality flags available to discriminate the AOD utilized. Confidence flags are also provided, with the interpolated data being lowest confidence and daytime AERONET constrained the highest. More details on the approach are available at https://mplnet.gsfc.nasa.gov/product-info/product_pages.cgi?p=AER. For this study with no lunar AOD available, the standard MPLNET aerosol data from night time are only those of lowest confidence quality.

GRASP approach on the other hand prioritizes photometric retrievals by selecting and accumulating lidar profiles in vicinity of the available almucantar observations, however without access to lunar AOD observations it relies on limiting time variation



of the retrieved columnar aerosol properties such as sphericity fraction, size distribution and chemical composition, to be able to treat lidar profiles during nighttime, when no co-incident sun-photometric observations are available. In addition, such limitation has to be fruitful in photometric retrievals close to noon, providing additional constraint for the observations that usually lack some information due to the narrower range of scattering angles.

### 3 Comparison strategy

The MPLNET data (profiles of extinction and backscatter and columnar AOD and LR at 532 nm) were selected for the same times as GRASP retrievals, that during day time are driven by almucantar measurement times, and at night are performed at 17:00, 20:00, 23:00 and 2:00 UTC. It should be outlined that MPLNET processing uses the 20 minutes cloud screened averages to provide profile for each minute, that are comparable with 30 minutes profile averaging performed for GRASP retrievals, mentioned in Section 2. The selected profiles are compared in a bin-to-bin manner guaranteeing comparison of the values provided for the same altitudes. Profiles that had less than of 35 vertical bins in the altitude range of interest in the MPLNET V3 L1.5 AER product were omitted completely from the comparison. Additional quality assurance on GRASP provided products could be applied in order to exclude values, that correspond to the retrievals that didn't achieve expected levels of measurements observations accuracies (see Table 1). Overall, 1904 out of 6450 for scenario 1 and 972 out of 4380 for scenario 2 profiles are quality assured from successfully processed combined MPLNET and sun-photometer data, corresponding to the filtering rates of 30% and 22%.

### 4 Daylight properties comparison

The comparison of aerosol properties provided by GRASP and MPLNET products are organized in the following way: a separate analysis is performed for daytime and nighttime properties estimations, within each diurnal group both columnar properties as well as their vertical distributions will be analyzed. It should be additionally outlined that MPLNET product has several levels of confidence, and by using the same daytime data as GRASP that corresponds to the time of combined AOD + almucantar observations, we select the data with no AOD time interpolation and therefore operating with the best quality data that are sun-photometer constrained.

At the same time since no lunar AOD measurements were available at KAUST AERONET site, the MPLNET data selected for comparison during nighttime will rely on "long-term" interpolation of AOD and therefore be the least assured. Meanwhile, GRASP nighttime retrievals rely on smoothness restrictions of time variation of columnar microphysical properties of aerosol, which propose a rather comparable yet different approach to constrain nighttime liar retrievals.



## 4.1 Comparison of columnar properties

For adequate comparison of GRASP and MPLNET aerosol products it is reasonable to start with the comparison of columnar properties, notably AOD and LR at 532 nm. Both of these values are not included into the state vector describing the aerosol properties that is optimized during GRASP retrievals (Lopatin et al., 2013, 2021) and both AOD and LR values are estimated on the base of physical model including retrieved size distribution, sphericity and chemical composition. The estimations of columnar values of AOD and LR are crucial in GRASP estimations of profiles of vertical distributions of aerosol extinction

and backscatter, which are performed in the following manner:

$$\sigma(\lambda, h) = \sum_{i=1}^{N} \tau_i(\lambda) v_i(h), \qquad (4)$$

$$\beta(\lambda, h) = \sum_{i=1}^{N} \frac{\tau_i(\lambda) v_i(h)}{S_i(\lambda)}, \qquad (5)$$

where $N$ denotes the number of aerosol modes, $\tau_i(\lambda)$ denotes aerosol optical depth of the corresponding mode, and $S_i$ lidar

ratio of the corresponding aerosol mode at 532, defined by Eq. 3 and $v_i(h)$ normalized aerosol vertical distribution profile of the corresponding mode. Since normalized vertical distribution profiles $v_i(h)$ are mostly influenced by lidar observations (Lopatin et al., 2013), and overall will be formed by the normalized NRB profile, having the same columnar values for AOD and LR is crucial for GRASP to provide similar to MPLNET profiles of extinction and backscatter.

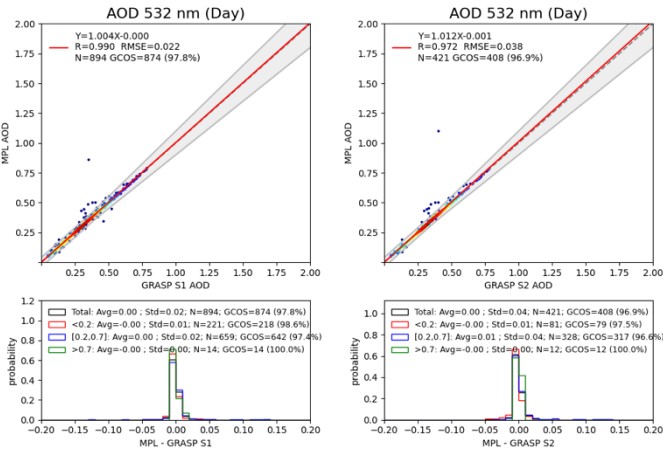

**Figure 2: Comparison of daytime columnar aerosol optical depth at 532 nm retrieved by GRASP and MPLNET over KAUST observation site for the period 2019-2022, for Scenario 1 (left) and Scenario 2 (right) GRASP retrievals.**

It should be noted, that Eqs. 4 and 5 can be used in a situation when aerosol model is represented by several modes, e.g., fine and coarse, like in scenario 2 retrievals analyzed in this study. Figure 2 shows the results of comparison of day-time AOD estimations provided by GRASP and MPLNET, for two types of GRASP retrievals excluding and including the volume

depolarization data provided by MPLNET. The comparison is exceptionally good due to the fact that during daytime MPLNET retrieval utilizes AOD observations as constraints. Since GRASP inverts combined AOD and almucantar data, the time





windows selected for the comparison should contain same AOD observations performed by sun-photometer. The observable differences are caused by the differences in estimation of AOD at 532 nm which is not directly observed by the sun-photometer. In the case of MPLNET it is done by second order polynomial fitting of spectral AOD observed in case of standard AERONET

sun-photometer at 440, 679, 870 and 1020 nm, while GRASP relies on the common physical model, including aerosol chemical composition that satisfies all types of the observations included in the retrieval (see Table 1 for details). Despite of these differences the statistical properties of the comparisons are exceptionally good, with correlation coefficients of 0.990 and 0.972 for Scenario 1 and 2 correspondingly, RMSE of 0.022 and 0.038, close to 1 slope and no biases at different ranges of AOD values.

The slight differences between scenario 1 and 2 are most likely related to the differences in the dataset used for the comparison, as additional requirements to the volume depolarization profiles exclude some of the data from processing using scenario 2 which nonetheless could be present in scenario 1. Such additional filtering may solely be responsible for the improvement of comparison statistics by excluding the low-quality data that still could be present in NRB profiles, making the retrievals more accurate. Another possibility is a higher flexibility of the aerosol model used in scenario 2, distinguishing properties of fine

and coarse aerosol particles and therefore operating with a bigger, and therefore, more flexible set of parameters during the retrievals, allowing to perform them more accurately. Overall, 91.28% and 89.78% of GRASP quality assured daytime AODs are laying within the error intervals provided in the aerosol MPLNET L1.5 V3 product.

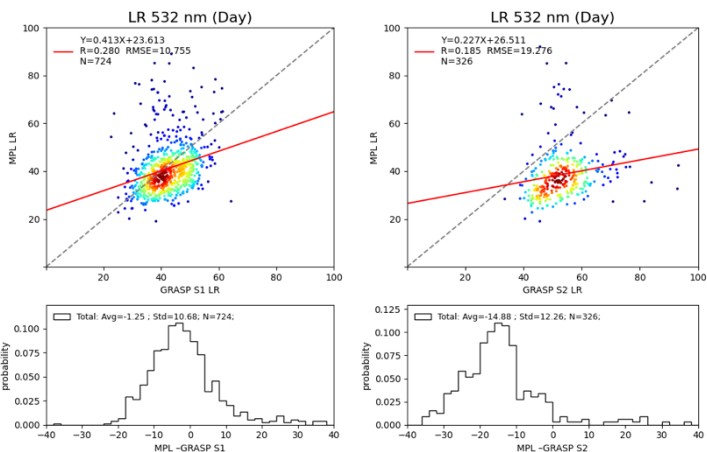






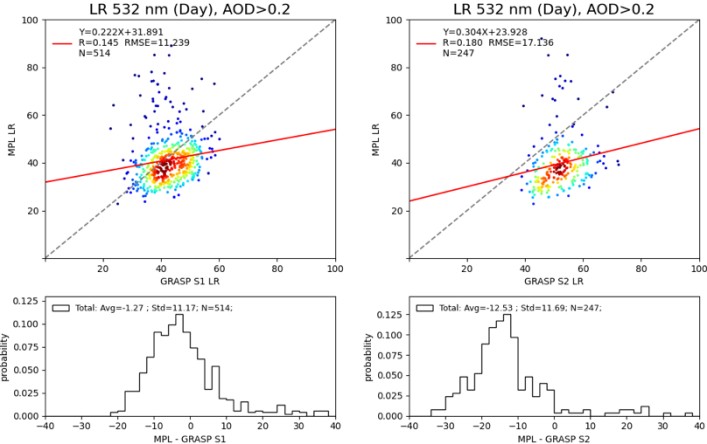

**Figure 3: Comparison of daytime columnar lidar ratio at 532 nm retrieved by GRASP and MPLNET over KAUST observation site for the period of 2019-2022, for Scenario 1 (left) and Scenario 2 (right) GRASP retrievals, including (top) and omitting (bottom) low AODs<0.2.**

Figure 3 shows results of comparison of day-time LR estimations provided by GRASP and MPLNET, for two types of GRASP retrievals excluding and including the volume depolarization data provided by MPLNET. Similar to AOD comparison in Fig. 2 the LR values are selected for the moments when AOD measurements were provided, guaranteeing therefore the best quality for MPLNET LR estimations. Nonetheless, the results of the comparison demonstrate lower statistical results, as those for AOD due to the different approaches used by MPLNET and GRASP to estimate LR. While MPLNET uses modified Fernald (Welton et al., 2002, Marenco et al., 1997, Fernald et al., 1984) algorithm with AOD-provided calibration to effectively

estimate columnar AOD and LR from the NRB lidar signal, GRASP relies on both angular dependencies of aerosol properties provided by almucantar observations together with normalized attenuated backscatter and, in case of scenario 2, volume depolarization profiles provided by MPLNET to estimate columnar microphysical properties of aerosol that are then be used to estimate the LR (Dubovik and King, 2000; Dubovik et al., 2006). It should be additionally outlined that in case of Scenario 2 GRASP operates with two columnar LR estimated for fine and coarse aerosol modes correspondingly, with an effective total

LR estimated following Eq. 3. The denominator in Eq. 3 in case of several aerosol modes could be estimated as follows (Dubovik et al., 2011):

$$\omega_0(\lambda) = \frac{\sum_{i=1}^{N} \omega_0^i(\lambda)\tau_i(\lambda)}{\sum_{i=1}^{N} \tau_i(\lambda)}, \tag{6}$$

$$P_{11}(\lambda, 180°) = \frac{\sum_{i=1}^{N} \omega_0^i(\lambda)\tau_i(\lambda)P_{11}^i(\lambda, 180°)}{\sum_{i=1}^{N} \tau_i(\lambda)}. \tag{7}$$

Following Fig. 3, both MPLNET and GRASP estimate LR at ~40±10 *Sr* which is within the ranges typical for desert dust (Welton et al., 2002, Muller at al., 2007, Schuster et al., 2012, Papayanis et al., 2008). It should be outlined that variability of retrieved LR is quite low, due to the dominance of desert dust usually present over the KAUST site. Certainly, this limited





variability range decreases the correlation and slope values. It should be mentioned, that KAUST observational site is located on the coast, and lower LR values as compared to desert dust, are most likely related to the marine aerosols influence.

In regards of Scenario 1 (left part of Fig. 3), both MPLNET and GRASP approaches are closer due to the similarity in aerosol assumptions, notably both methods operate with only one columnar value of LR. Nonetheless, MPLNET L1.5 estimations have a higher spread in this case reaching values up to 100 *Sr*. The reason of such high values is most likely due to lidar signal attenuation during daytime with high dust loading. In such cases the aerosol top height tends to be estimated too low if not screened properly, and thus aerosols are present at the start of the constrained backward Fernald inversion just above the layer

which is assumed to be molecular, resulting in high retrieval bias of the LR. Improved filtering of such cases is planned in L2 MPLNET processing.

GRASP demonstrate somewhat smaller spread as compared to MPLNET notably due to the use of physical model to estimate LR with the parameters additionally limited in temporal variation. In the case of scenario 2 a more pronounced discrepancy could be observed, with both methods demonstrating similar spread, and MPLNET data having a notable bias of -14.2 *Sr* as

compared to GRASP estimations. In scenario 2 GRASP estimations of LR are slightly higher, being ~50±10 *Sr*. These observed differences are most likely present due to the possibility of columnar LR variations due to the presence of second mode. Both scenarios demonstrate quite low RMSEs around 11 and 19 *Sr* correspondingly. Overall, 28.04% and 4.6% of GRASP quality assured daytime LRs for scenario 1 and Scenario2 correspondingly are laying within the error intervals provided in the aerosol MPLNET L1.5 V3 product.

Lower panel of Fig. 3 compares the retrievals of columnar LR for Scenarios 1 and 2 correspondingly, but with additional filtering that omits the retrieval cases with low AODs, below 0.2. Indeed, such filtering have proven to be very useful for comparing advanced aerosol products, such as Angström exponent or Single Scattering Albedo (SSA) (e.g., Wagner and Silva, 2008, Chen et al., 2020), allowing to effectively exclude cases with weak aerosol contribution to the atmospheric observation, and therefore with lower quality of retrievals. The filtered result show similar mean values of the LRs, with noticeably lowered

variation of GRASP values in Scenario 2. Scenario 2 also demonstrates lower RMSE after filtering, being 17.14 instead of 19 Sr, while scenario 1 remained almost the same (~11 Sr). The inability of such filtering to get rid of high LR outliers in MPLNET product and significantly lower the RMSE and MPLNET LR variation most likely indicates the signal attenuation and retrieval issues discussed above.






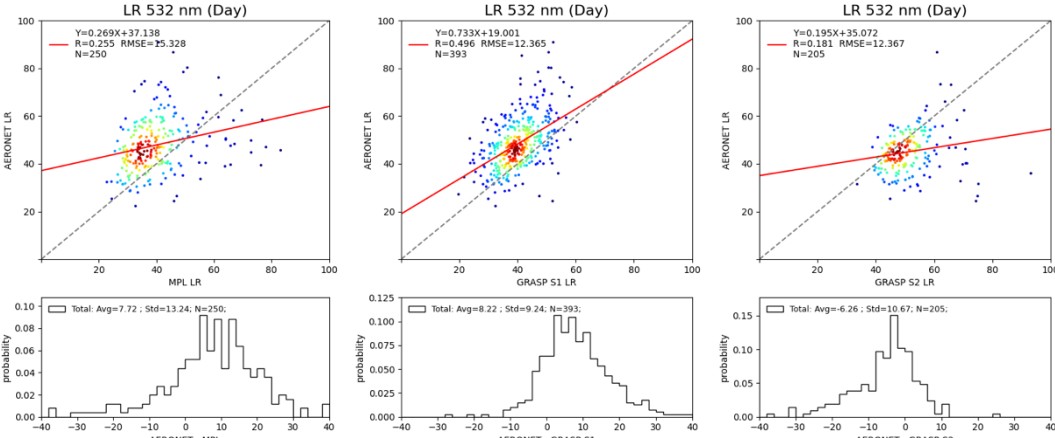

**Figure 4: Comparison of daytime columnar lidar ratio at 532 nm retrieved by GRASP and MPLNET and estimated from AERONET retrievals over KAUST observation site for the period of 2019-2022, AERONET vs MPLNET (left), AERONET vs GRASP Scenario 1 (middle) and Scenario 2 (right).**

Figure 4 compares columnar lidar ratio at 532 between 4 available products: GRASP Scenario1 and 2, MPLNET LR, and the

LR from standard AERONET processing. All four products provide close LR values with average bias not exceeding 10 Sr.

At the same time GRASP Scenario2 and AERONET estimated LR show the closest values, with a small underestimation (~6

Sr) from GRASP product. Overall AERONET estimated values are 45 ±10 Sr. At the same time Scenario 1 and MPLNET data

in relation to AERONET estimated values have comparable performance, with overestimation of ~7 and ~9 Sr for each product

correspondingly. These differences are in general agreement with Fig. 3 showing a bias between Scenario 1 and Scenario 2

products in relation to MPLNET. Generally, AERONET LR estimations provide a more reasonable values, given the expected

aerosol type over KAUST site is dust. In these regards usage of volume depolarization in Scenario 2 should have provided

additional sensitivity to dust properties leading to a more stable retrieval.

For the illustrative purposes it could be particularly interesting to look into the details of the GRASP/MPLNET diurnal

comparisons, e.g., during the time period of one or two days. It should be emphasized though, that analysis of one day period

could not provide a very profound analysis and can represent a particular case, that is not typical for the majority of

observations. Figure 5 shows an example of the time sequence of AOD's and lidar ratios provided by GRASP Scenario 1 and

2 and MPLNET retrievals for the period of 21 September 2022. The nighttime period is shadowed in blue, MPLNET

estimations are presented in black, while GRASP ones are plotted in green and red for scenarios 1 and 2 correspondingly,

AERONET provided estimations of AOD at 532 nm (interpolated from 440 and 670 nm Angström exponent) and LR

calculated on the base of retrieved microphysical properties (including, size distribution, complex refractive index and

spherical particles fraction) are shown in blue. It should be noted that MPLNET provides the data for each minute (with 20-

minute sliding window), while GRASP uses 15-minute lidar data accumulation (totaling a 30 minutes accumulation) around

available combined AOD/Almucantar measurements performed by AERONET during day time, and around 2:00, 20:00. and

23:00 UTC at nighttime (shown in rounds). As it can be clearly seen, similar to the right part of Fig. 2, the AOD daytime





comparison is exceptionally good between all four products. Such outcome is expected, since GRASP directly uses AOD
values to fit together with sky radiance and lidar data, at the same time MPLNET uses AOD provided by AERONET to
constrain their retrievals. Similar to Figs. 3 and 4 behavior of time evolution also could be observed for the lidar ratio
estimations. The estimations performed during the day time are closer than during nighttime, though a significant difference
could be observed between Scenario 1 and Scenario 2 covering a range of almost 30 Sr. Overall MPLNET and AERONET

LR estimations demonstrate higher variability with both AERONET and MPLNET values being overall closer to Scenario1
during daytime observations.

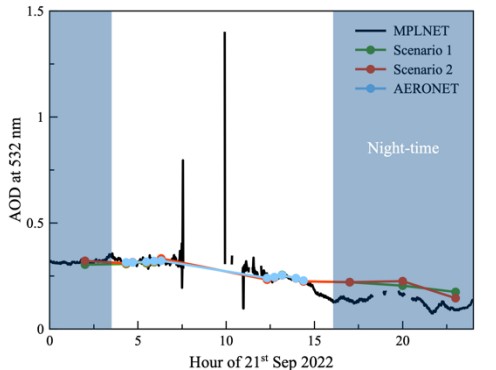 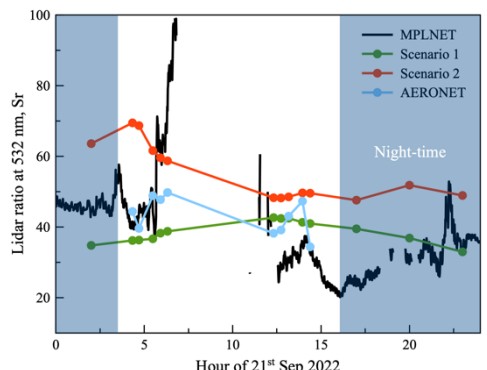

**Figure 5: Time sequence of the Aerosol optical depth (left), and lidar ratio (right) at 532 nm for 21 September 2022 for MPLNET(Black), GRASP Scenario1 (green) and Scenario 2 (red) and AERONET (blue) observations, AERONET AOD values are**
**interpolated at 532 nm using 440/675 nm Angström exponent.**

## 4.2 Comparison of vertical profiles

Figure 6 shows the results of layer-to-layer comparison of day-time estimations of vertical extinction profiles provided by
GRASP and MPLNET for two types of GRASP retrievals excluding and including the volume depolarization data provided
by MPLNET. Both methods show very good agreement with scenario 1 having a slightly better agreement due to the bigger

similarities between GRASP and MPLNET approaches, notably the use of only one aerosol mode distributed within a single
vertical profile. The correlation coefficients are 0.980 and 0.975 for scenario 1 and scenario 2, correspondingly. RMSEs are
very low, not exceeding 16 $Mm^{-1}$ and linear regression slopes exceptionally good, being 0.85 and 0.84 correspondingly.
Overall, both methods do not have significant biases against each other with these parameters no lower than -5.94 $Mm^{-1}$ and -
5.43 $Mm^{-1}$ for Scenario 1 and 2 correspondingly, with majority of differences located within the -50 to 25 $Mm^{-1}$ range. Overall,

85.73% and 85.42% of GRASP quality assured vertical extinction profile values are within the error margin provided for this
parameter in the aerosol MPLNET L1.5 V3 product for scenarios 1 and 2 correspondingly.

It is worth mentioning that following the diagram in Fig. 6, GRASP generally provides lower values for the extinction profiles
than the ones provided by MPLNET. The reason of this discrepancy may lay in the differences of the aerosol profile treatment
implied by both methods. Indeed, both methods provide a vertical profile of extinction whose integration provides almost

identical columnar AOD values (see Fig. 2). The main difference is the integration ranges, and extra assumptions made to




perform it. MPLNET lidar signals are provided from 250 m above ground to 30 km in the NRB products for all instruments at 532 nm (the lower limit was higher for older instruments and fixed at 527 m). MPLNET aerosol processing first determines the aerosol top height as described above, and the bottom of the calibration zone serves as the upper range limit for aerosol retrievals. The bottom limit is the surface, and lidar signals below 250 m are filled in as a constant using the signal value just

above 250 m. GRASP at the same time extrapolates the aerosol profile outside the range of 270 — 5670 m by assuming aerosol to be constant from lower limit to the ground level, and linearly decreasing up to the altitudes of 40km starting from the upper limit (Lopatin et al., 2013). Thus, a comparison within the limited altitude range leaves some parts of a wider bottom-to-top profile behind, effectively lowering this part of the profile, since in GRASP retrievals the omitted parts still contribute to the columnar AODs, that may not be fully accounted by MPLNET.

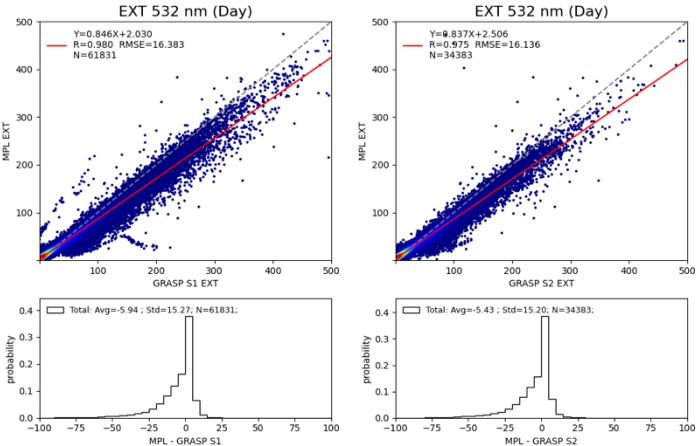


**Figure 6: Layer-to-layer comparison of daytime aerosol vertical extinction profiles at 532 nm retrieved by GRASP and estimated by MPLNET over KAUST observation site for the period of 2019-2022, for Scenario 1 (left) and Scenario 2 (right) GRASP retrievals.**

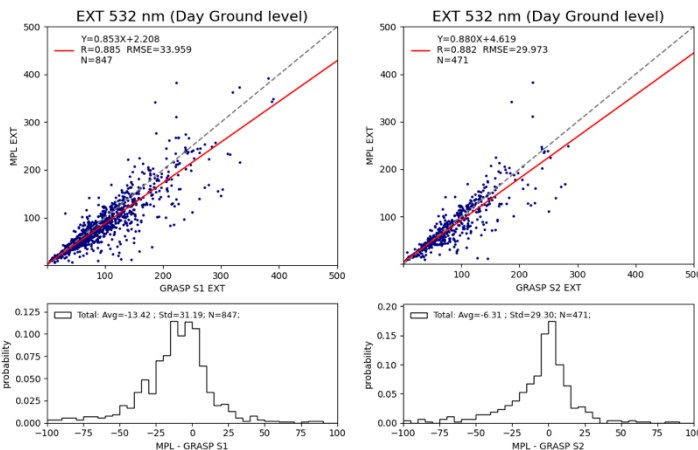


**Figure 7: Comparison of daytime aerosol extinction at 532 nm at ground level estimated by GRASP and MPLNET over KAUST observation site for the period of 2019-2022, for Scenario 1 (left) and Scenario 2 (right) GRASP retrievals.**



In order to clarify the above aspect, an additional comparison of the aerosol extinction estimated at the ground layer was performed in order to investigate how well an assumption of constant aerosol distribution or lidar signal at lower layers, notably in lidar cut-off zone, affects the estimation of this value. Such comparison is reasonable to perform for daytime observations,

where both columnar AOD and vertical extinction profiles demonstrate outstanding agreement (see Figs. 2 and 6), thus allowing to limit the influence of other factors that affect the estimates of the ground level extinction to the differences between approaches used to estimate extinction in both products.

Figure 7 illustrates the comparison of aerosol daytime extinction at 532 nm in the lowest altitude layer provided by GRASP and MPLNET products, located approximately at 50 m above sea level. It should be outlined that these values are not supported

by lidar observations, but rather estimated using constrain of total columnar AOD and assumptions described above. Following the general logic of the comparison, GRASP products excluding and including information on volume depolarization provided by MPLNET are presented. As could be seen in Fig. 7 a rather simple assumption on aerosol distribution made in GRASP still allows to estimate ground level extinctions rather accurately, with RMSEs not exceeding 38.3 and 43.5 $Mm^{-1}$, correlation coefficients of 0.89 and 0.88, impressive linear regression slopes of 0.85 and 0.88 and average biases not exceeding -13.4 and

-6.3 $Mm^{-1}$ for scenarios 1 and 2 correspondingly. It should be outlined that Scenario 2 uses two vertical distribution profiles separated between fine and coarse modes, which provides it an additional flexibility in describing the total aerosol extinction at the ground level, which may explain why the values of slope and bias are better in the case of comparison with scenario 2.

Figure 8 shows results of layer-to-layer comparison of day-time estimations of vertical backscatter profiles provided by GRASP and MPLNET, for two types of GRASP retrievals excluding and including the volume depolarization data provided

by MPLNET. Both methods show very good agreement with scenario 1 having slightly better agreement due to the bigger similarities between GRASP and MPLNET approaches, notably the use of only one aerosol mode distributed within single vertical profile. The correlation coefficients are 0.96 and 0.95 for scenario 1 and scenario 2 correspondingly, RMSEs are very low not exceeding 0.53 and 0.8 $Sr^{-1} \cdot Mm^{-1}$ correspondingly, while linear regression slopes are 0.8 for Scenario 1 and being slightly lower (0.65) for scenario 2. This most likely is related to the differences in columnar LR estimations discussed above,

additionally, the presence of the second vertical profile, providing a more detailed distribution of GRASP scenario 2 LR vertically as compared to MPLNET retrievals may impact the comparison. Scenario 1 has very low negative bias (-0.18 $Sr^{-1} \cdot Mm^{-1}$), following the trends of extinction profile and columnar LR estimations (see Eq. 5). A low (-0.41 $Sr^{-1} \cdot Mm^{-1}$), but observable bias is present in scenario 2, similarly to scenario 1 propagating into the backscattering estimations from vertical extinction profiles and columnar LR comparison differences.





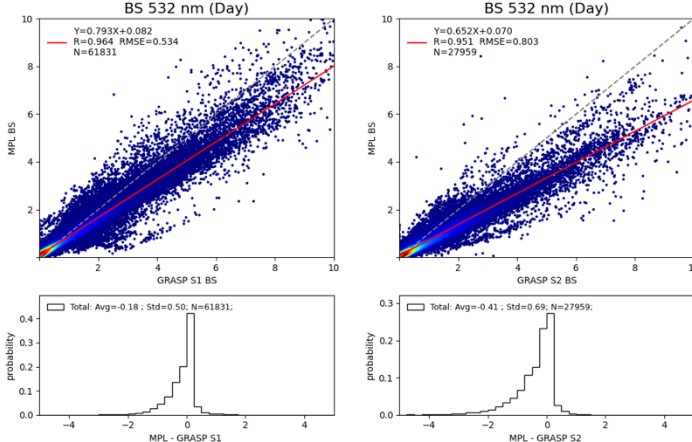


**Figure 8: Layer-to-layer comparison of daytime aerosol vertical backscatter profiles at 532 nm retrieved by GRASP and estimated by MPLNET over KAUST observation site for the period of 2019-2022, for Scenario 1 (left) and Scenario 2 (right) GRASP retrievals.**

Overall, 80.60% and 69.31% of GRASP quality assured daytime vertical backscatter profile values are within the error margin provided for this parameter in the aerosol MPLNET L1.5 V3 product for scenarios 1 and 2 correspondingly.

Similar to the Figs. 6 and 8, the daytime comparison of profiles of aerosol extinction and backscatter provided for GARSP scenario1 and 2 and MPLNET at 532 nm for the 21 September 2022 shown in Fig. 9 are very encouraging. Small biases that could be observed in backscatter profiles are due to the differences in lidar ratio estimations (see the right panel of Fig. 5) used in different scenarios of GRASP and MPLNET L1.5 retrievals. Significant differences could be observed sometimes in the lower part of the profiles, which are located in the cut-off zone of the MPL lidar, which, however, do not demonstrate big

significance in the overall comparison for the ground-based concentration levels shown in Fig. 7. It should be outlined that for this particular case the signal top cut-off in GRASP and MPLNET treatment is slightly different, with MPLNET reaching 6oo0 m altitudes. This creates some discrepancy in the estimation of extinction at the highest altitude (~5700 m) between both GRASP products and MPLNET profiles. Since the value of the top layer is extrapolated to the TOA, this may cause some observable bias between different products with AOD values nonetheless being exactly the same (see e.g., Fig. 5). Indeed,

MPLNET, unlike GRASP, allows lidar signal top cut-off to vary with time, and similar approach will be applied to GRASP processing of MPLNET data, to avoid such discrepancies in the future.





**Figure 9: Comparison of profiles of aerosol extinction (top) and backscatter (bottom) at 532 nm retrieved by GRASP for Scenario 1**
**(green) and Scenario 2 (red) with provided by MPLNET aerosol product (black) for the daytime period of 21 September 2022.**

Additionally, it should be emphasized, that unlike the majority of the comparison cases presented in Fig. 6, where very little difference could be observed between estimations provided by both scenarios, scenario 2 for the case on 21 September demonstrates better accordance with MPLNET provided profile.

## 5 Nighttime properties comparison

This section presents the comparisons of retrieved columnar and vertical properties of aerosol from MPLNET and GRASP during nighttime. It should be additionally outlined that during nighttime both methods do not rely on any photometric observations due to lack of lunar AOD at KAUST site, and use completely different methods to estimate the values of aerosol properties. Without lunar AOD from AERONET, nighttime MPLNET estimations are performed from lidar observations only and do not rely on any spectral interpolation as compared to daytime retrievals, being the least assured data in the MPLNET



V3 L1.5 dataset. GRASP on the other hand estimates columnar aerosol properties due to a combination of consecutive lidar observations combined with sun-photometric measurements performed during daytime under an assumption of limited change of aerosol columnar properties over time (see Lopatin et al., 2021 for details). As a matter of fact, KAUST observation site that is dominated by one aerosol type and provides quite stable temporal aerosol load (Parajuli et al., 2020) is more than suitable for the retrievals under such assumptions. However, the multi-temporal approach used in GRASP is not limited only

to stable aerosol situations, as was demonstrated by Lopatin et al., 2021.

## 5.1 Comparison of columnar properties

Figure 10 shows the results of comparison of night-time AOD estimations provided by GRASP and MPLNET, for GRASP retrievals following scenarios 1 and 2, respectively. The comparison is less convincing as compared to the daytime retrievals (see Fig. 2). At the same time, taking into account that during nighttime both methods do not rely on any AOD observations

as compared to the day time and, overall, use completely different methods to estimate the AOD values this comparison is more than encouraging.

Despite of these differences the statistical properties of the comparisons are inspiring, with correlation coefficients of 0.53 and 0.62, RMSE of 0.282 and 0.22, slope values of 0.62 and 0.85 for scenarios 1 and 2 correspondingly. Total biases are low, 0 and 0.05 respectively, with the same biases at low AOD (<0.2) of 0.08. The slight differences between scenario 1 and 2 are

most likely related to the differences in the dataset used for the comparison, as additional requirements to the volume depolarization profiles exclude some of the data from processing using scenario 2 which nonetheless could be present in scenario 1. Such additional filtering may be responsible for the improvement of comparison statistics by excluding low quality data that could still be present in NRB profiles, making retrievals of scenario 1 less accurate. Another possibility is a higher flexibility of the aerosol model used in scenario 2, distinguishing properties of fine and coarse aerosol particles and therefore

operating a more flexible set of retrieval parameters, allowing more accurate retrievals. Overall, 4.97% and 5.97% of GRASP quality assured nighttime AODs for scenario 1 and scenario 2 correspondingly are laying within the error intervals provided in the aerosol MPLNET L1.5 V3 product.

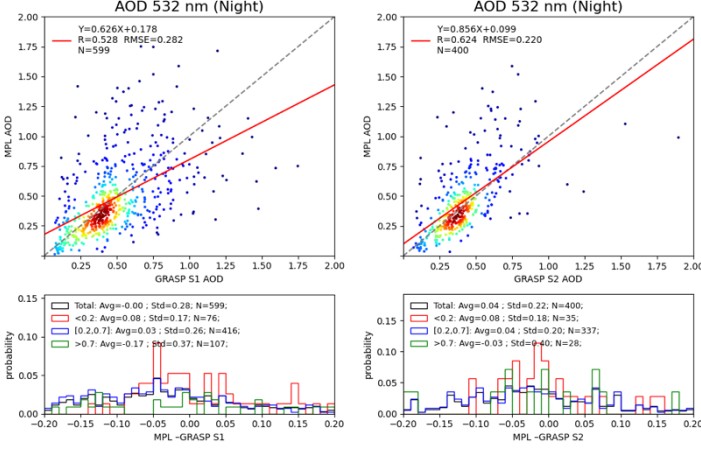





**Figure 10: Comparison of nighttime columnar aerosol optical depth at 532 nm retrieved by GRASP and MPLNET over KAUST**
**observation site for the period 2019-2022 over KAUST observation site for the period of 2019-2022, for scenario 1 (left) and scenario 2 (right) GRASP retrievals.**

Figure 11 shows the results of comparison of nighttime LR estimations provided by GRASP and MPLNET, for two types of GRASP retrievals excluding and including the volume depolarization data provided by MPLNET. Similar to daytime, both
MPLNET and GRASP estimate LR at ~40±10 *Sr* which is within the typical ranges for desert dust. As already noticed above, the variability of retrieved LR is quite low due to the dominance of desert dust that projects to the lower correlation and less stable slope values, rendering linear fit metrics to be less helpful than in other cases presented.

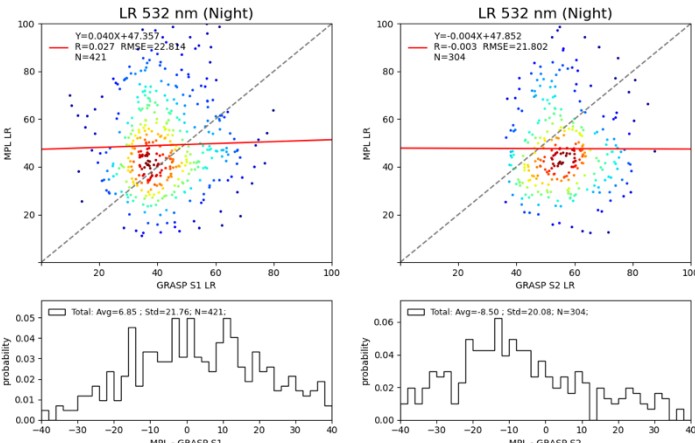

**Figure 11: Comparison of nighttime columnar lidar ratio (left) at 532 nm retrieved by GRASP and MPLNET over KAUST**
**observation site for the period 2019-2022 during, for scenario 1 (left) and scenario 2 (right) GRASP retrievals.**

In general, the nighttime statistics of columnar LR comparison at 532 nm is very similar to the daytime (see Fig. 3). For scenario1 (left part of Fig. 8) both MPLNET and GRASP approaches are closer due to the similarity in aerosol assumptions. Both scenarios demonstrate a wider spread as compared to daytime retrievals. Consecutively, both scenarios demonstrate higher RMSEs as compared to daytime around 22.8 and 12.3 *Sr* correspondingly. Also, a bigger discrepancy could be observed,
with both methods demonstrating similar spread, and MPLNET data having a notable bias of -7, and 9 *Sr* correspondingly as compared to GRASP estimations from both scenarios. Similar to daytime, in scenario 2 GRASP estimations of LR are slightly higher, being ~50±10 *Sr*. These observed differences are most likely present due to the possibility of columnar LR variations due to the presence of second mode. Overall, 7.8% and 7.18% of GRASP quality assured nighttime LRs for scenario1 and 2 correspondingly are laying within the error intervals provided in the aerosol MPLNET L1.5 V3 product.


A more detailed analysis, performed for the nighttime period of 21 September 2021 could be seen in blue shadowed areas of Fig. 5. The AOD comparisons, similarly to the overall ones, shown in Fig. 10, are quite encouraging. While this is expected for the daytime data, since the AOD measurements are included in the GRASP retrievals, it is not the case for the nighttime,





where no AOD data were used. At the same time an observable bias (up to ~0.1) in nighttime AOD estimations could be seen
between scenario1 and scenario 2, which gets higher in the middle of the night. Its presence is explained by the additional
restrictions on columnar AOD provided from the necessity to fit volume depolarization profiles, which may make the
smoothness restrictions applied on aerosol concentration less important for scenario 2. It also could be observed that GRASP
AOD estimations for both scenarios, being restricted by time variability, are quite smooth, while the data provided by MPLNET
(derived from lidar observations, as indicated) undergoes significant variations, most likely due to the time interpolation
methods that are used to provide lidar calibration in between the available AOD observations provided by sun-photometer.
Probably, since no lunar AOD is available to stabilize the temporal interpolation, these assumptions may accumulate significant
errors overnight. Similar behavior of time evolution is also observed for the lidar ratio estimations. While estimations
performed during the day time are close, some significant differences may be observed during the night.

**5.2 Comparison of vertical profiles**

Figure 12 shows results of layer-to-layer comparison of nighttime estimations of vertical extinction profiles provided by
GRASP and MPLNET, for two types of GRASP retrievals excluding and including the volume depolarization data provided
by MPLNET. Both methods show good agreement with scenario 1 having slightly better agreement due to the bigger
similarities between GRASP and MPLNET approaches, notably the use of only one aerosol mode distributed within single
vertical profile. The correlation coefficients are 0.774 and 0.784 for scenario 1 and scenario 2 correspondingly, with similar
RMSEs not exceeding 43 $Mm^{-1}$ and linear regression slopes are quite good, 0.65 for Scenario 1 and being slightly higher (0.70)
for scenario 2. This most likely could be explained by the presence of the second vertical profile, providing a more detailed
distribution of aerosol vertically as compared to MPLNET retrievals. Scenario 1 has a negative bias (-7.65 $Mm^{-1}$), following
the trends of extinction profile and columnar LR estimations (see Eq. 5). Even lower (-5.37 $Mm^{-1}$) bias is present in scenario
2 as compared to scenario 1. As compared to daytime retrievals (Fig. 3) vertical profiles of extinction show less agreement,
which most likely propagates from nighttime AOD retrieval uncertainties through Eq. 4. Overall, 43.60% and 43.25% of
GRASP quality assured nighttime vertical extinction profile values are within the error margin provided for this parameter in
the aerosol MPLNET L1.5 V3 product for scenarios 1 and 2 correspondingly.





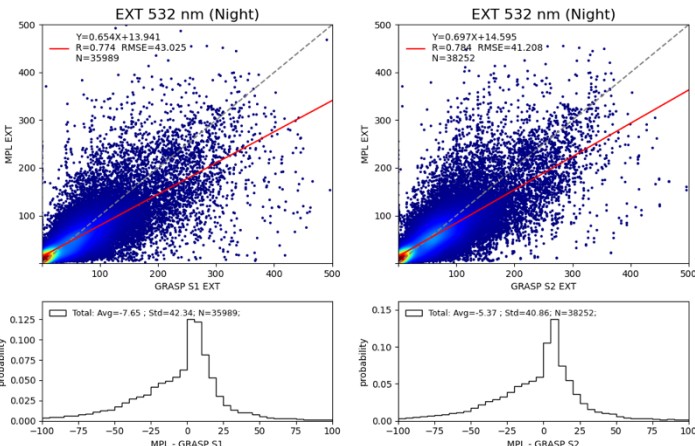

**Figure 12: Layer-to-layer comparison of nighttime aerosol vertical extinction profiles at 532 nm retrieved by GRASP and estimated**
**by MPLNET for scenario 1 (left) and scenario 2 (right) GRASP retrievals.**

Figure 13 shows results of layer-to-layer comparison of nighttime estimations of vertical backscatter profiles provided by GRASP and MPLNET, for two types of GRASP retrievals excluding and including the volume depolarization data provided by MPLNET. Both methods show good agreement with scenario 2 having a slightly better one. The correlation coefficients are 0.67 and 0.79 for scenario 1 and scenario 2 correspondingly, RMSEs are very low, not exceeding 1.2 and 0.8 $Sr^{-1} \cdot Mm^{-1}$

correspondingly and linear regression slopes are quite good, 0.84 for scenario 1 and being slightly lower (0.64) for scenario 2. Most likely the presence of the second vertical profile, providing a more detailed distribution of LR vertically (following Eqs. 4 and 5) as compared to MPLNET retrievals explains the observable differences between two scenarios. Scenario 1 has very low negative bias (-0.12 $Sr^{-1} \cdot Mm^{-1}$), following the trends of extinction profile and columnar LR estimations (see Eq. 5). A low (-0.30 $Sr^{-1} \cdot Mm^{-1}$), but observable bias is present in scenario 2, similarly to daytime retrievals propagating into the

backscattering estimations from vertical extinction profiles and columnar LR estimation differences (see Fig. 11), following Eq. 5.

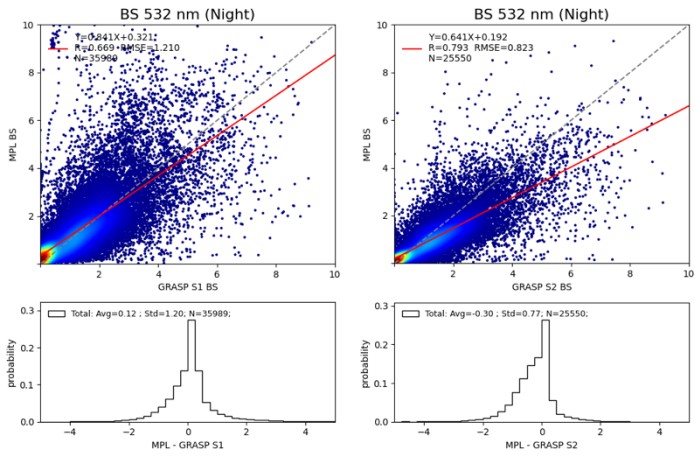






Overall, 44.28% and 46.84% of GRASP quality assured nighttime vertical backscatter profile values are within the error margin provided for this parameter in the aerosol MPLNET L1.5 V3 product for scenarios 1 and 2 correspondingly.

**Figure 14: Comparison of profiles of aerosol extinction (top) and backscatter (bottom) at 532 nm retrieved by GRASP for scenario 1 (green) and scenario 2 (red) with provided by MPLNET aerosol product (black) for the nighttime period of 21 September 2022.**


Detailed analysis of nighttime profiles for the 21 September 2022, presented in Fig. 14 similarly to overall comparison presented in Figs. 12 and 13 shows high similarities between both scenarios GRASP and MPLNET products, showing almost exact correspondence (e.g., 17:00, 20:00 and 23:00) to slightly different profiles' magnitudes with similar shapes (e.g., 02:00). Such discrepancies evidently propagate from the differences in the estimation of AOD and lidar ratios at 532 nm that are

provided by different scenarios of GRASP and MPLNET products. As can be seen in Fig. 5, even when nighttime AOD values from all three products demonstrate very close values, LR undergoes some significant shifts for all of the observations on 21



September 2022. Overall, both AOD and LR biases between scenarios 1 and 2 of GRASP products directly propagate in the backscatter profiles, taking into account that the analyzed case was dominated by coarse particles, meaning that influence of separated fine mode in scenario 2 has limited significance. Nonetheless, for this particular period nighttime retrievals

demonstrate better agreement both in extinction and backscatter than a daytime comparison shown in Fig. 9. However, it should be outlined, that such behavior is not typical, as demonstrated by Figs. 6,8 and 12,13 correspondingly.

Table 3 summarizes the comparison results for GRASP and MPLNET retrieved parameters for columnar AOD, LR, and vertical extinction and backscatter at 532 nm retrieved during day and night, for GRASP retrievals excluding and including the volume depolarization data.


**Table 3: Summary of the comparison results for GRASP and MPLNET retrieved columnar AOD, LR, and vertical extinction and backscatter at 532 nm retrieved during day and night time, for scenario 1 and 2 of GRASP retrievals.**

| Parameter /Variable | Daytime | | Nighttime | |
|---|---|---|---|---|
| | Scenario1 | Scenario2 | Scenario1 | Scenario2 |
| Columnar AOD | | | | |
| R | 0.990 | 0.972 | 0.528 | 0.624 |
| RMSE | 0.022 | 0.038 | 0.282 | 0.220 |
| BIAS | 0.0 | 0.0 | 0.0 | 0.04 |
| Slope | 1.00 | 1.01 | 0.63 | 0.86 |
| Columnar lidar ratio | | | | |
| R | 0.280 | 0.185 | 0.027 | 0.003 |
| RMSE, Sr | 10.76 | 19.27 | 22.81 | 21.80 |
| BIAS, Sr | -1.3 | -14.9 | 6.85 | -8.50 |
| Slope | 0.41 | 0.23 | 0.04 | 0.004 |
| Vertical profile of extinction | | | | |
| R | 0.980 | 0.975 | 0.774 | 0.784 |
| RMSE, $Mm^{-1}$ | 16.38 | 16.14 | 43.03 | 41.21 |
| BIAS, $Mm^{-1}$ | -5.94 | -5.43 | -7.65 | -5.37 |
| Slope | 0.85 | 0.84 | 0.654 | 0.697 |
| Vertical profile of backscatter | | | | |
| R | 0.964 | 0.951 | 0.669 | 0.793 |
| RMSE, $Sr^{-1} Mm^{-1}$ | 0.53 | 0.80 | 1.21 | 0.82 |
| BIAS, $Sr^{-1} Mm^{-1}$ | -0.18 | -0.41 | 0.12 | -0.30 |
| Slope | 0.8 | 0.65 | 0.84 | 0.64 |

## 6 Seasonal diurnal analysis of aerosol properties over KAUST site in 2020-2022

This section focuses on analyzing the differences in aerosol properties that are retrieved during day and nighttime over KAUST site during the observation period used in this study (March 2019 — December 2022). Scenario 2 data was used for the



analysis, since it provides the most complex aerosol modelling, allowing to separate aerosol vertical profiles into fine and coarse modes (See Table 2 for details) due to inclusion of vertical profiles of volume depolarizations provided by MPLNET into the retrieval, it is expected to provide the most detailed description of aerosol properties available (as compared to scenario

660 1).

Figure 15 presents the comparison of the diurnal median aerosol fine, coarse spherical and non-spherical fractions, estimated for four seasons. These components generally could be associated with anthropogenic activities (fine), maritime aerosols (coarse spherical) and desert dust (coarse non-spherical).

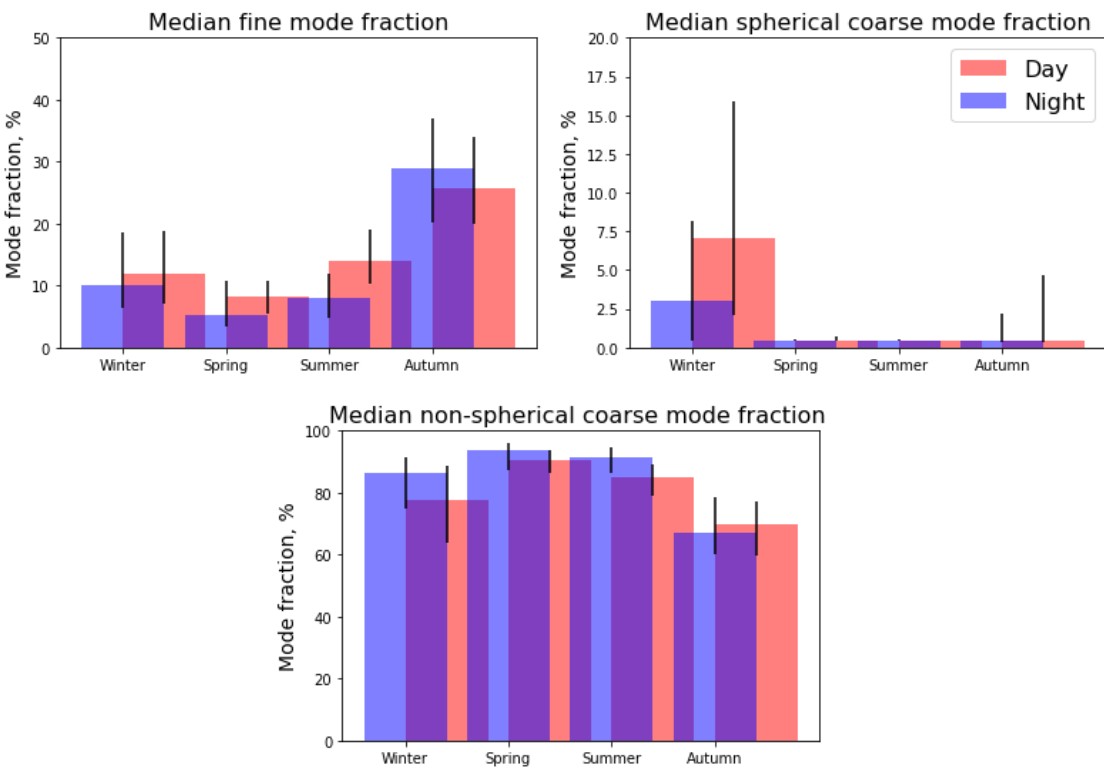


**Figure 15: Median aerosol fine (top left), coarse spherical (top right) and non-spherical (bottom) component fractions for daytime (red) and nighttime (blue) retrievals for winter, spring, summer and autumn seasons. The corresponding 25% -75% percentile variations are shown in black.**

Figure 15 clearly shows significant variation in aerosol composition from season to season, e.g., winter season demonstrates a

much higher coarse spherical contribution, while autumn has a more pronounced fine mode. It should be outlined that all seasons apart of winter are dominated by coarse non-spherical particles that may be associated with a more significant presence of desert dust. Additionally, in winter a more significant diurnal variation both in spherical and non-spherical components could be observed, at the same time fine mode shows a much lower difference between day and night. Such behavior is most likely related with local aerosol transport events, notably the prevailing winds, that bring more maritime particles during the

day.




Both spring and summer seasons could be described as "dusty" period, indicating a higher non-spherical particles concentration (>90% by volume) as compared to autumn and winter. During this period a more significant variation in the fine mode could be observed, especially in summer, with higher fine particles load during the day, indicating most probably a contribution from human activity.

Figure16 showing diurnal median complex refractive index at 532 nm for four seasons additionally supports the conclusions described above. For e.g., real part of the complex refractive index of coarse aerosol component (spherical and non-spherical components are not distinguished by refractive index and have same values) indicates lower indices in winter (both real and imaginary), which is reasonable for mixtures of maritime and dust aerosols. It should be noted, that in autumn real part of refractive index has similar values, however imaginary part suggests stronger absorption than in winter and similar to Spring-

Summer, while the fraction of coarse spherical mode remains very low (see Fig. 15), most likely indicating changes in microphysical properties of desert dust.

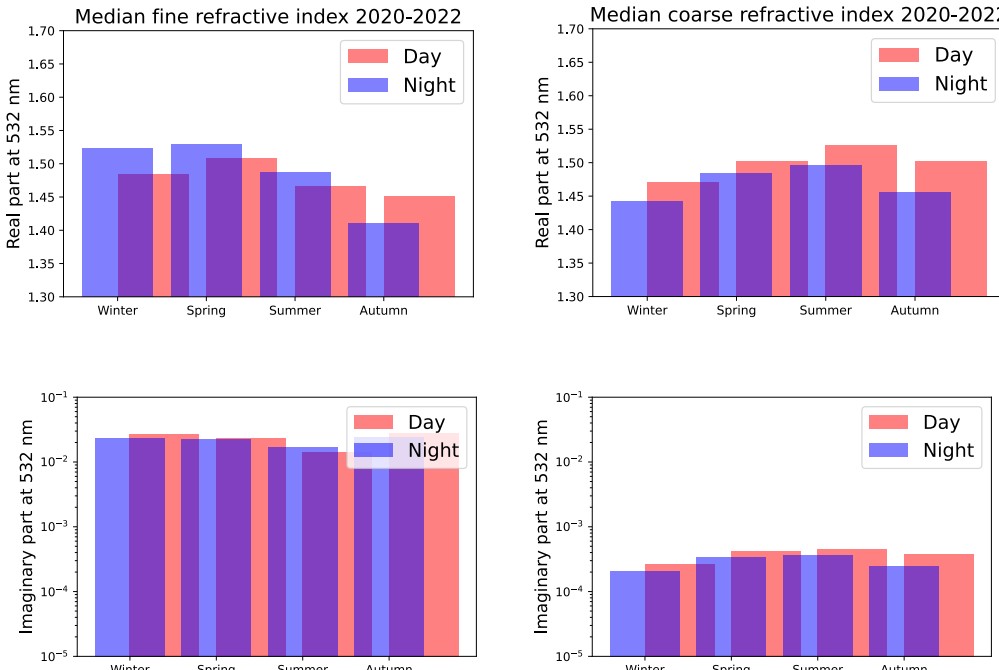

**Figure 16: Median complex refractive index at 532 nm for fine (left) and combined coarse spherical and non-spherical (right) aerosol**
**components for daytime (red) and nighttime (blue) retrievals for winter, spring, summer and autumn seasons, real part presented at the top imaginary at the bottom.**

In overall, fine mode refractive index in the left panel of Fig. 16 shows stronger absorption than coarse component, indicating particles of different chemical composition, that are most likely related to human activities. At the same time, it should be emphasized that discrimination of fine and coarse mode refractive indices in a generally coarse-dominated (see Fig. 15)

environment remains a challenging task.



Figure 17 demonstrates median vertical profiles of fine, coarse spherical and non-spherical components for daytime and nighttime for four seasons. Generally, all component profiles are following the same trends as the columnar compositions, e.g., showing little to no spherical particles in Spring-Autumn. At the same time, it could be observed that in winter and autumn all aerosol components are generally located lower (usually below 3.5 km) than in summer and spring, while all components
appear to be well mixed.

Similar to the columnar properties in Fig. 15 a diurnal cycle of notable increase of coarse spherical particles concertation with corresponding decrease of non-spherical one during daytime in winter could be observed (top middle panel of Fig. 17), additionally, it could be outlined that the biggest change appears in the lower part of atmosphere, below 2 km.

During the spring and summer, a significant diurnal variation of fine component (left panels of Fig. 17) could also
be observed, notably at altitude layers close to the ground (below 500 m), while in Winter-Autumn these layers appear to be more elevated (~1 Km) with overall increase at nighttime, indicating most likely seasonal diurnal shift in anthropogenic activities.

Coarse non-spherical component profiles (right panels in Fig. 17) have a noticeable maximum around 1km both during day- and nighttime in all seasons except Summer, when the layers appear to be well mixed up to the maximum observation altitudes.
This peak has a slight decrease in winter and autumn during the nighttime. A significant diurnal cycle of coarse non-spherical component in the layers above 3km could be observed in autumn, which could be associated with the change of prevailing winds at these altitudes, introducing more maritime particles.

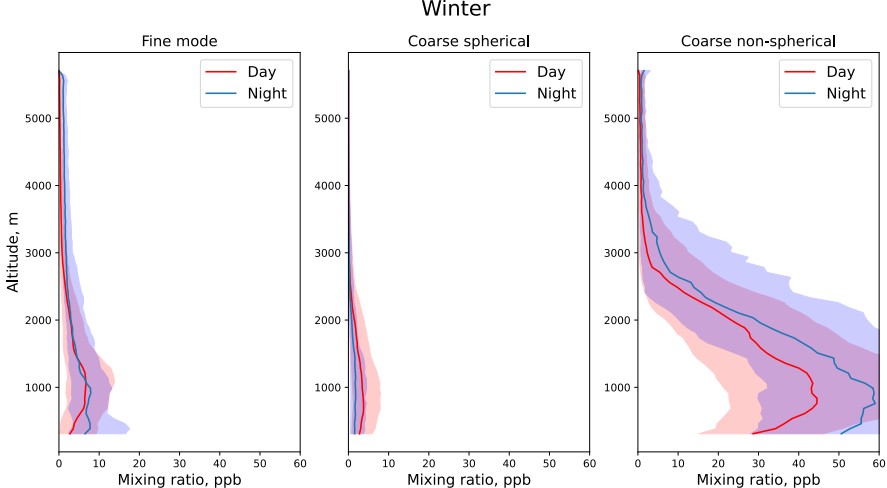



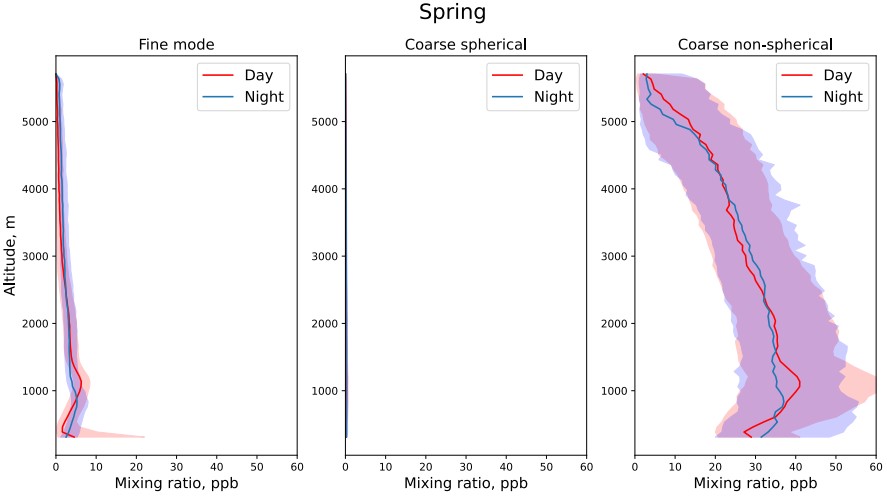

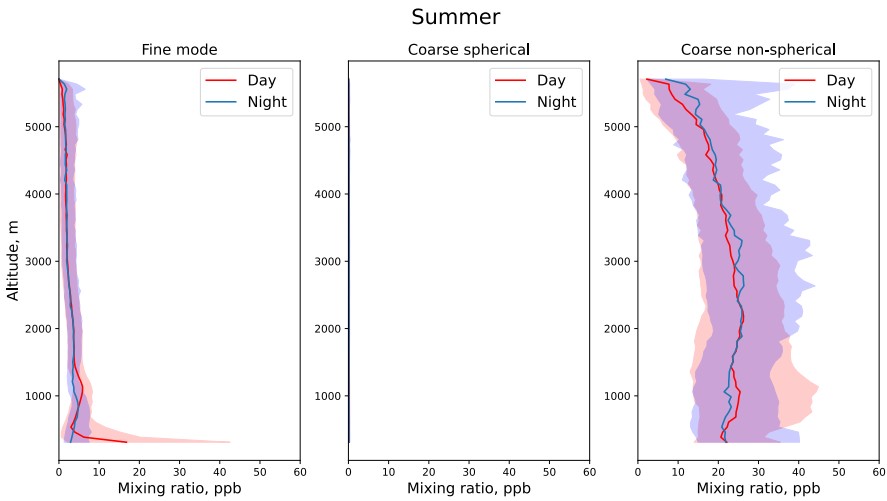




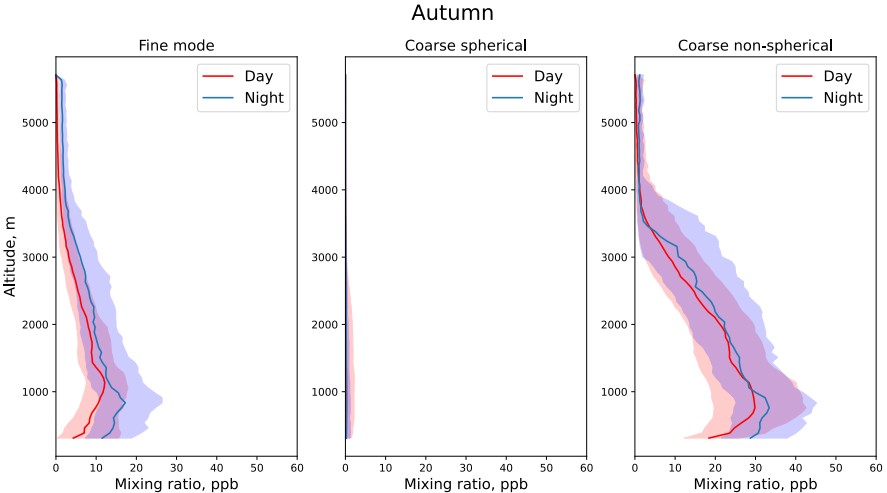

**Figure 17: Median vertical profiles of fine (left), coarse spherical (middle) and coarse non-spherical (right) components for daytime (red) and nighttime (blue) retrievals for (top to bottom) winter, spring, summer and autumn seasons. The corresponding 25% - 75% percentile variations are shown in shadows of corresponding colors.**

## 7 Conclusions

Almost three consecutive years of data starting from march 2019 till December 2022 collected over KAUST observation site including vertical profiles of volume depolarization provided by MPLNET lidar were processed using the GRASP software under assumption of limited time variability of such columnar properties as size distribution, chemical composition and sphericity fraction. As the result of the processing, columnar optical properties such as AOD and lidar ratio together with vertical profiles of extinction and backscatter at 532 nm were estimated for the retrievals excluding and including volume depolarization data provided by MPLNET.

The resulting properties were co-located with MPLNET V3 L1.5 aerosol product and compared. Additional emphasis was made on separating day- and night- time retrievals as well as on potential benefits of utilizing volume depolarization profiles in the retrievals.

In overall, both columnar and vertical MPLNET and GRASP products demonstrated a better agreement for daytime retrievals excluding the depolarization information. Such outcome is rather expected, as in scenario 1 GRASP and MPLNET share more of assumptions, as compared to scenario 2. It should be additionally outlined that both products demonstrate lower columnar LR ratios as would be expected to such dust dominated site as KAUST as well as compared to AERONET estimations.

In overall the following results for daytime retrievals without accounting for polarization profiles were achieved:

- For columnar AOD, correlation coefficient of 0.99, RMSE of 0.022, and 0.0 total bias including bias at low (<0.2) AOD, and linear regression slope of 1.

- Columnar LR correlation coefficient of 0.282, RMSE of 10.76 *Sr*, bias of -1.3 *Sr* and linear regression slope of 0.41;



- Vertical profiles of extinction correlation coefficient of 0.98, RMSE of 16.38 $Mm^{-1}$, total bias of -5.94 $Mm^{-1}$ and linear regression slope of 0.85;

- Vertical profiles of backscatter correlation coefficient of 0.964, RMSE of 0.53 $Sr^{-1} \cdot Mm^{-1}$, total bias of -0.18 $Sr^{-1} \cdot Mm^{-1}$ and linear regression slope of 0.8.

Inclusion of volume depolarization profiles in the GRASP retrievals allows to distinguish between columnar properties and vertical distribution of fine and coarse aerosol modes, thus providing a more complex model and diverting further from the

assumptions implied in MPLNET retrievals (e.g., same lidar ratio for all observed atmospheric layers). At the same time in dust dominated cases these differences are not expected to have a strong impact on the retrievals. Meanwhile the presence of the volume depolarization profiles causes a significant difference in columnar LR estimations, thus limiting the agreement between MPLNET and GRASP products. At the same time scenario 2 product demonstrates LR ratios as would be expected to such dust dominated site as KAUST, additionally they are closer to AERONET estimations than scenario 1.

In overall, the following results for daytime retrievals accounting for polarization profiles were achieved:
- For columnar AOD, correlation coefficient of 0.972, RMSE of 0.038, and 0.0 total bias including bias at low (<0.2) AOD, and linear regression slope of 1.01
- Columnar LR correlation coefficient of 0.185, RMSE of 19.27 $Sr$, total bias of -14.9 $Sr$ and linear regression slope of 0.23;

- Vertical profiles of extinction correlation coefficient of 0.975, RMSE of 16.14 $Mm^{-1}$, total bias of -5.43 $Mm^{-1}$ and linear regression slope of 0.84;
- Vertical profiles of backscatter correlation coefficient of 0.951, RMSE of 0.80 $Sr^{-1} \cdot Mm^{-1}$, total bias of -0.41 $Sr^{-1} \cdot Mm^{-1}$ and linear regression slope of 0.65.

Additional comparison performed for estimated values of daytime extinction profiles at a ground level were performed in order to assess the impact of assumptions of constant aerosol vertical distribution in the cut off zone of lidar observations implied in GRASP. Estimations provided by GRASP for retrievals including the volume depolarization profiles, demonstrated slightly better linear regression slope and bias with comparable correlation coefficients and RMSE, most notably due to a higher flexibility allowing to describe the total ground level extinction as a sum of the values of fine and coarse aerosol modes.


The comparison of properties retrieved during nighttime is expectedly worse as compared to the daytime retrievals. During nighttime both methods did not rely on any photometric observations due to lack of lunar AOD at this site, and use completely different methods to estimate the values of aerosol properties. Nighttime MPLNET estimations were performed from lidar observations only and do not rely on any spectral interpolation as compared to daytime retrievals, GRASP on the other hand

estimates columnar aerosol properties due to a combination of consecutive lidar observations combined with sun-photometric measurements performed during daytime under an assumption of limited change of aerosol columnar properties over time.



Despite of these differences the statistical properties of the comparisons are encouraging and the following results for nighttime retrievals without accounting for polarization profiles were achieved:

- For columnar AOD, correlation coefficient of 0.528, RMSE of 0.282, 0.0 total bias, including bias at low (<0.2) AOD of 0.09, and linear regression slope of 0.63;
- Columnar LR correlation coefficient of 0.027, RMSE of 22.81 $Sr$, total bias of 6.85 $Sr$ and linear regression slope of 0.04;
- Vertical profiles of extinction correlation coefficient of 0.774, RMSE of 43.03 $Mm^{-1}$, total bias of -7.65 $Mm^{-1}$ and linear regression slope of 0.654;
- Vertical profiles of backscatter correlation coefficient of 0.669, RMSE of 1.21 $Sr^{-1} \cdot Mm^{-1}$, total bias of 0.12 $Sr^{-1} \cdot Mm^{-1}$ and linear regression slope of 0.84.

What concerns nighttime retrievals accounting for polarization profiles, in overall the following results were achieved:

- For columnar AOD, correlation coefficient of 0.624, RMSE of 0.220, 0.04 total bias, including bias at low (<0.2) AOD of 0.08, and linear regression slope of 0.86;
- Columnar LR correlation coefficient of 0.003, RMSE of 21.80 $Sr$, total bias of -6.85 $Sr$ and linear regression slope of 0.04;
- Vertical profiles of extinction correlation coefficient of 0.784, RMSE of 41.21 $Mm^{-1}$, total bias of -5.37 $Mm^{-1}$ and linear regression slope of 0.697;
- Vertical profiles of backscatter correlation coefficient of 0.793, RMSE of 0.82 $Sr^{-1} \cdot Mm^{-1}$, total bias of -0.30 $Sr^{-1} \cdot Mm^{-1}$ and linear regression slope of 0.64.

Inclusion of the volume depolarization observations had observable influence on the agreement between MPLNET and GRASP estimated values, both columnar and vertical for both night- and daytime values. The strongest difference was observed in columnar LR estimations with retrievals of scenario 2 having a noticeable positive bias against GRASP scenario 1, MPLNET and AERONET estimated values. Biased values belong to a range that is expected for desert dust particles, a primary aerosol component over the KAUST observation site. However, a decisive conclusion on the improvements of accounting for depolarization data on nighttime retrievals would require additional studies. Those should include independent nighttime observations of aerosol columnar and vertical properties, e.g., lunar photometry and/or Raman lidars allowing to precisely evaluate estimations of nighttime aerosol properties provided by both methods and thus accurately estimate the impact of polarization data inclusion on GRASP products.

Analysis of statistical distribution of columnar and vertical aerosol properties distinguished between fine, coarse spherical and coarse non-spherical aerosol components suggests noticeable changes in aerosol diurnal and seasonal variability.



**Competing interests**

At least one of the co-authors is a member of the editorial board of AMT.

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
