# Peer review of "Comparison of diurnal aerosol products retrieved from combinations of micro-pulse lidar and sun-photometer over KAUST observation site"

_EGUsphere, 2024_

## Referee Comment (RC2)

**Review of "Comparison of diurnal aerosol products retrieved from combinations of micro-pulse lidar and sun-photometer over KAUST observation site" by Lopatin et al.**

The manuscript *"Comparison of diurnal aerosol products retrieved from combinations of micro-pulse lidar and sun-photometer over KAUST observation site"* presents and discusses an almost 4-year comparison between retrievals of columnar aerosol optical depth (AOD), lidar ratio (LR) and particle backscatter ($b_p$) and extinction ($a_p$) coefficient profiles at 532nm, using the MPLNET and GRASP retrieval approaches over a coastal, desert dust dominated site in the Arabian Peninsula (KAUST). Both MPLNET and GRASP utilize collocated AERONET and lidar data, however different approaches in handling of these data. Specifically, MPLNET operational retrieval uses AERONET AOD linearly interpolated at 532nm as a constrain to apply the Klett-Fernald inversion to derive $a_p$ and $b_p$, while GRASP uses spectral AERONET AOD as an input to the retrieval together with spectral solar almucantar data and lidar signals at 532nm (from MPLNET) thus inverting the sun-photometer and lidar datasets together. Both algorithms are in need of specific assumptions regarding either the temporal changes in AOD (in case of MPLNET) or temporal changes in particles microphysical properties (in case of GRASP) during night-time due to the absence of sun-photometer data. As such, the comparisons are separately performed during daytime and night-time. Further, two scenarios are demonstrated where the volume linear depolarization ratio (VLDR) at 532nm provided by MPLNET is included in the GRASP retrieval (scenario 2) or not (scenario 1). The paper also demonstrates the seasonal diurnal variability of the complex refractive index, concentration vertical distribution and relative contribution of fine, coarse spherical and coarse non-spherical aerosols to the atmospheric column over KAUST site for the overall period of the comparisons, offering an in-sight to the aerosol patterns in the area but also possibly to relation with meteorological conditions (i.e. wind direction).

I believe the study falls well within the scope of AMT. The manuscript is well-written and structured, the presentation of the methodology is thorough and the quality of the figures high. Furthermore, the authors give credit to related work and the results support the conclusions.

However, in order to help improving the manuscript, I would kindly suggest the authors to take into account the following specific comments.

1. Throughout the manuscript the authors refer to the use of the "AERONET data", however it is never mentioned which level or version of AERONET datasets is utilized. Although for MPLNET retrievals it is implied that Level 2 AERONET data are used (since it is stated that Level 2 MPLNET retrievals are produced after Level 2 AERONET data are available), this is not obvious for GRASP. I believe it would be helpful for the reader if the authors provide a few more details on the Level of data used in GRASP but also specific files names, list of parameters, units (if applicable) etc.

2. It is mentioned in several cases in the manuscript that the use of VLDR data in GRASP results in worse comparison with MPLNET predominantly during daytime observations when AERONET data are also included in the retrieval. The authors argue that this could be a result of several reasons including the assumption of a single aerosol mode in the atmospheric column when VLDR data are not utilized. Is it possible that this result for daytime retrievals, could also be related with the non-spherical particles treatment in GRASP? More specifically, could the fact that VLDR is used as an input, highlight the shortcomings of the spheroid assumption for dust particles shape in this case when the algorithm tries to balance the fitting of both AOD and VLDR?

3. Assuming an operational use of GRASP for long-term lidar/sun-photometer measurements' processing (i.e. for any MPLNET site), could there be a threshold for the VLDR values to differentiate between cases when it is useful for this parameter to be included in the retrieval or not? The results for the specific site mostly refer to desert dust dominated cases and indeed the use of VLDR seems to 'push' the retrieval of LR at 532nm closer to the values expected form

the literature. However, when the amount of depolarizing aerosols in the atmospheric column is very low, would the use of VLDR still be beneficial?

4. **Line 324:** "lidar" instead of "liar"

5. **Line 360:** It I not clear to me here and in other places in the manuscript what you mean with *'a bigger, and therefore, more flexible set of parameters during the retrievals, allowing to perform them more accurately'*. Wouldn't scenario 2 predominately benefit the retrieval of the coarse mode non-spherical fraction? However, the comparisons shown are between total LR, AOD, $a_p$ and $b_p$ values.

6. **Line 385:** '*LR at ~40±10 Sr which is within the ranges typical for desert dust*'. I believe here you mean at 532nm.

7. **Line 533:** *'scenario 2 for the case on 21 September demonstrates better accordance with MPLNET provided profile'*. I believe it seems from figures 7, 8 and 9 that this is mostly true for $a_p$ and not $b_p$. In relation to question 2, do you think that this could be also related with the spheroidal assumption which would mostly affect the $b_p$?

---

## Author Comment (AC1)

Reply to the anonymous referee #2 review on **Comparison of diurnal aerosol products retrieved from combinations of micro-pulse lidar and sun-photometer over KAUST observation site" by Lopatin et al.**

Authors would like to express our gratitude for the time and effort reviewers had dedicated to reviewing the article. We believe that the suggestions made have significantly helped us to improve the quality of the manuscript

Below we provide detailed answers to the specific comments of the reviewers:

*1. Throughout the manuscript the authors refer to the use of the "AERONET data", however it is never mentioned which level or version of AERONET datasets is utilized. Although for MPLNET retrievals it is implied that Level 2 AERONET data are used (since it is stated that Level 2 MPLNET retrievals are produced after Level 2 AERONET data are available), this is not obvious for GRASP. I believe it would be helpful for the reader if the authors provide a few more details on the Level of data used in GRASP but also specific files names, list of parameters, units (if applicable) etc.*

Answer 1: "**RAW almucantar**" AERONET data, available at https://aeronet.gsfc.nasa.gov/cgi-bin/webtoolinv_v3
In combination with "**Total Optical Depth based on AOD Level**" available at https://aeronet.gsfc.nasa.gov/cgi-bin/webtool_aod_v3 both of Level1.5 were used. Clarifications were added to the text:
**Lines 165–168**: "We have processed almost three consecutive years of data starting from march 2019 till December 2022 collected over KAUST observation site including vertical profiles of volume depolarization provided by MPLNET lidar in combination with co-located AERONET observations, notably the L1.5 total optical thickness and raw almucantars available at (https://aeronet.gsfc.nasa.gov/cgi-bin/webtool_aod_v3 and https://aeronet.gsfc.nasa.gov/cgi-bin/webtool_inv_v3 correspondingly) using the version 1.1.1 of GRASP software."

*2. It is mentioned in several cases in the manuscript that the use of VLDR data in GRASP results in worse comparison with MPLNET predominantly during daytime observations when AERONET data are also included in the retrieval. The authors argue that this could be a result of several reasons including the assumption of a single aerosol mode in the atmospheric column when VLDR data are not utilized. Is it possible that this result for daytime retrievals, could also be related with the non-spherical particles treatment in GRASP? More specifically, could the fact that VLDR is used as an input, highlight the shortcomings of the spheroid assumption for dust particles shape in this case when the algorithm tries to balance the fitting of both AOD and VLDR?*

Answer 2: Authors would like to thank the reviewer on pointing out the lack of discussion of such important subject. Nonetheless it should be outlined that, unfortunately, this issue is not very straightforward to address within the scope of the study, and definitely an additional study should be conducted in order to properly address the issue. Below, we have provided detailed explanations, together with additions to the article text, that were also linked with question 7.

Indeed, while the Inclusion of VLDR data results in the more realistic lidar ratio retrievals it also causes the appearance of larger differences between extinction and backscatter obtained by GRASP and MPLNET approaches. The authors consider that most probably both of these effects can be explained by the fact that adding VLDR increases overall the information content of inverted data and helps to improve the overall retrieval. At the same time, this positive impact can only be expected if newly added VLDR data are interpreted correctly. In this regard choice of particle shape models is critical for interpretation VLDR data. The randomly oriented spheroids model used here is the most popular model used in many aerosol retrieval algorithms (see the text below), because of two reasons. First, it captures well many features of scattering by non-spherical particles. Second, it essentially is the only model that can be used in quantitative retrieval. On the other hand, the accuracy of this model is often questioned because spheroid is the simplest non-spherical particle that doesn't have some realistic features inherent for real dust particles, e.g. sharp edges. Therefore, questioning the potential drawbacks of this model in our studies is fully reasonable. In these regards, full verification of spheroid model accuracy is clearly out the scope of current paper, while the analysis of the possible manifestations of spheroid model is evidently desirable and it has been done. Specifically, we can highlight that both AOD and sky-radiances in Almucantar obtained from passive measurements together with VLDR and attenuated backscatter

observations obtained from active lidar observation are fitted within the expected accuracy levels within a wide angular and spectral range. Such good reproduction of all inverted data could hardly be achieved if seriously erroneous scattering model would be used.

For improving the clarity of the paper, the following discussion was added to Section 4.1:

**lines: 421–436**: "It should be outlined, that it is expected for all methods to demonstrate better agreement in daytime extinction profiles. Indeed, all methods are constrained by AOD observations both for GRASP and MPLNET. At the same time, backscatter profiles comparison results will rely on how close the estimations of columnar lidar ratios provided by these methods are. The differences in LR estimations observed in Fig. 3 could have different origins. For example, GRASP S2 used VLDR data that provide additional information that is expected to improve the retrievals. However, the accurate interpretation of VLDR requires the reliable model of non-spherical aerosol scattering properties. The approach of randomly oriented spheroids developed by Dubovik et al., (2006) has been used in all four methods (GRASP S1 and S2, MPLNET and AERONET). While spheroids mixture is evidently an idealistic model, it had shown to be efficient in many applications for quantitative characterisation of intensity and polarization scattering properties of non-spherical particles in wide angular and spectral ranges. Indeed, spheroids have been successfully employed in passive AERONET ground-based (Dubovik et al., 2006) and spaceborne multi-angular polarimeters (Dubovik et al., 2011, 2021; Hasekamp et al., 2024); to extensive complex data sets of observations including in-situ (Espinosa et al, 2017, 2019; Bazo et al., 2024), as well as to active measurements (e.g., Lopatin et al., 2021) and to various combinations of active and passive remote sensing observations both ground-based (e.g., Lopatin et al., 2013, 2021) and spaceborne (Xu et al., 2021). At the same time, MPLNET products used in this study do not rely on volume depolarisation observations, and hence provide extinction and backscatter profiles together with columnar lidar ratio without any relation to spheroid model."

*3. Assuming an operational use of GRASP for long-term lidar/sun-photometer measurements' processing (i.e. for any MPLNET site), could there be a threshold for the VLDR values to differentiate between cases when it is useful for this parameter to be included in the retrieval or not? The results for the specific site mostly refer to desert dust dominated cases and indeed the use of VLDR seems to 'push' the retrieval of LR at 532nm closer to the values expected form the literature. However, when the amount of depolarizing aerosols in the atmospheric column is very low, would the use of VLDR still be beneficial?*

Answer3: For the inversion procedure low or close to 0 VLDR provides as much useful information as elevated values, assuming that VLDR is provided with the claimed accuracy. On the other hand, if VLDR values are not reliable due to the low signal-to-noise ratio, no useful information could be extracted. Indeed, such situation may require establishing some thresholds for filtering out the low-quality data. In fact, such operations are already performed by the MPLNET team providing L15 data. Such filtering is one of the main reasons why Scenario 1 and 2 retrievals deal with the different number of quality assured observations available for inversions within the given periods.

For improving clarity of the paper, the following passage was added to the manuscript:

**Lines 245 254**: "Indeed, inclusion of additional observations into the retrieval can bring additional benefits only in the cases when there is sufficient information on aerosol properties. In these regards, both high and low volume depolarisation ratios could be useful providing either information on properties of coarse non-spherical particles or spherical particles (both fine and coarse) correspondingly. However, low volume depolarisation ratio lidar observations can suffer from a significantly lower signal to noise ratios for the values of VLDR close to 0. Therefore, quality assurance on volume depolarisation provided by MPLNET L15 NRB data allows exclusion of such cases from the retrieval, assuring high quality extended retrievals even in cases without significant dust loads."
**lines 363–365**: "The slight differences between Scenarios 1 and 2 are most likely related to the differences in the dataset used for the comparison, as additional requirements to the volume depolarization profiles exclude some of the data from processing using scenario 2 which nonetheless could be present in Scenario 1, 4380 against 6450 correspondingly."

*4. Line 324: "lidar" instead of "liar*

Answer 4: corrected

*5. **Line 360:** It I not clear to me here and in other places in the manuscript what you mean with 'a bigger, and therefore, more flexible set of parameters during the retrievals, allowing to perform them more accurately'. Wouldn't scenario 2 predominately benefit the retrieval of the coarse mode non-spherical fraction? However, the comparisons shown are between total LR, AOD, ap and bp values.*

Answer 5: Scenario 2 effectively doubles the number of parameters that describe aerosol vertical distribution, this as compared to the scenario 1 increases the flexibility to fit lidar observations more accurately. Indeed, the backscatter and VLDR at each altitude are modelled by two values (one for each fine and coarse modes), instead of one (total) in S1, which makes it easier to find out a combination that fits observation data than in the case with the model that depends only on one parameter. Comparisons of fits are performed for all approaches, while only GRASP S2 distinguishes between fine and coarse modes.

For improving clarity of the paper, the following clarifications was added to the manuscript:

**lines 371–377:** "Another possibility is the use of the aerosol model with higher flexibility in Scenario 2 that distinguishes the properties of fine and coarse aerosol particles has a doubled number of parameters that can be used to reproduce observations compared to Scenario 1. Specifically, in Scenario 2 the properties of aerosol at each layer depends on the concentrations of fine and coarse particles, while LR is fixed for each fraction for the entire column, at the same time LRs for fine and coarse are different. In Scenario 1 lidar signal is fit by the model that retrieve LR and values of total aerosol at each layer. Therefore, Scenario 2 operates with a more flexible aerosol model and allows us to fit lidar observations more accurately under the same total AOD constraints."

*6. **Line 385:** 'LR at ~40±10 Sr which is within the ranges typical for desert dust'. I believe here you mean at 532nm.*

Answer 6: wavelength was added to the passage, **line 396**.

*7. **Line 533:** 'scenario 2 for the case on 21 September demonstrates better accordance with MPLNET provided profile'. I believe it seems from figures 7, 8 and 9 that this is mostly true for ap and not bp. In relation to question 2, do you think that this could be also related with the spheroidal assumption which would mostly affect the bp?*

Answer 7: In the additional discussion provided in answer 2 we would like to emphasize that use of spheroidal model is not the only factor that may affect the observed differences between the result obtained by MPLNET and by both GRASP S1 and S2. For example, the following procedures may introduce the differences: the vertical profile extrapolation above and below lidar observation range, the restrictions on temporal variability of columnar aerosol properties used in GRASP S1 and S2, the separation of vertical profiles of fine and coarse aerosol modes in GARSP S2, etc. Thus, it is very challenging to isolate the possible impact of spheroid model limitations on the retrievals. Additional study, that includes supplemental model-independent data, e.g. lidar ratios estimated by Raman lidar retrievals, may be conducted in the future to properly address this question.

The following discussion was added to the text:
**Lines 548–567**:
"Comparing Figs. 8 and 4., representing the estimations of total columnar lidar ratio by 4 different methods (AERONET, MPLNET, GARSP S1 and S2), it is clearly seen that once the extinction is constrained by AOD, the main difference in profiles originates from total columnar LR estimation difference, propagating into the backscatter profiles. As discussed in Section 4.1 there are several differences in the modelling approach between the four methods. Three of them (AERONET, GARSP S1 and S2) utilise the same spheroidal particle assumption to model non-spherical scattering, which provides a rather broad range of total columnar lidar ratios to reproduce diverse observations (Dubovik et al., 2006, Lopatin et al., 2021). For example, same as in similar studies by Lopatin et al., 2021 spheroid model had demonstrated ability to provide adequate fits all available observations data on 21 September 2022 within the expected accuracy of each observation (see Table 1). Those included: - spectral AOD (daily average residual 0.0018 and 0.0035 for S1 and S2 correspondingly), - sky-radiances in almucantars (daily average residual 4.44% and 4.46% for S1 and S2 correspondingly), - attenuated backscatter (daily average residual 1.32% and 3.42% for S1 and S2 correspondingly) and volume depolarisation at 532 nm (daily average residual

0.9% for S2) and derived LR 532 values closer to literature expectations for desert dust as compared to the LR used in MPNET retrievals. In addition, it should be noted, that the impact of utilising the spheroidal aerosol model could not be isolated from other factors that significantly influence the retrievals. Namely, in difference with MPLNET approach, GRASP accounts for aerosol in total atmospheric column, not only in the part observed by lidar (e.g., see inconsistencies in profile estimations above 6000m in Fig. 8), and in GRASP Scenario 2 aerosol is represented by two aerosol modes. In addition, the temporal restrictions on variability of aerosol columnar properties do not allow sharp temporal variations of LR in both GARSP retrievals, that, in a contrast, could be observed in MPLNET retrievals (see e.g., ~6:00 and ~12:00 values in the right panel of Fig. 4). Such analysis remains out of the scope of this study, and could be performed in the future should additional data (e.g., coincident LR retrievals from Raman lidars) become available."

---

## Author Comment (AC2)

Reply to the Gregory Schuster (referee #1) review on **Comparison of diurnal aerosol products retrieved from combinations of micro-pulse lidar and sun-photometer over KAUST observation site" by Lopatin et al.**

Authors would like to express our gratitude for the time and effort reviewers had dedicated to reviewing the article. We believe that the suggestions made have significantly helped us to improve the quality of the manuscript

Below we provide detailed answers to the specific comments of the reviewers:

*Thus, I would like to suggest that the authors replace some of the scatterplots with boxplots (or mean & standard error plots), and re-tool the analysis towards a student's T-test or similar. For instance, it would be rather easy to turn Figures 3&4 into a single figure with 7 notched box-and-whisker plots (one box each for MPL_LR, S1_LR, S2_LR, MPL_LR(AOD>0.2), S1_LR(AOD>0.2), S2_LR(AOD>0.2), and AERONET_LR. Each box is essentially a mini-histogram, so you'll have a nice visual of 7 histograms right next to one another for easy comparison (or add Fig 11 to obtain 10 boxplots). Do the notches overlap, indicating statistical agreement? Or do the medians have large separations? Are all of the boxes about the same size (indicating similar spreads), or are some larger than others? I think that this would be a much more enjoyable and useful way to look at the data than scatter plots that are nearly spherical. This is not a requirement, but I think that you'll retain more reader interest if you make this change.*

Answer: Authors are very greatful for such a suggestion, which certainly greatly improves the transparency of LR analysis. Figures 3 and 4 were replaced with one boxplot for the daytime comparison (both including filtering by AOD 0.2 and omiting one). Former Figure 11 (new Figure 10) also replaced with a boxplot, summarizing the nightitme retrievals. Discussions corresponding to figures and their analysiswere were re-done as suggested.

*The authors do a nice job of showing day/night and seasonal variations of the complex refractive index in Fig 16, but why not do the same thing with lidar ratio? Some of their LR discussion already suggested that seasonal variation in the sea salt / dust partitioning was causing differences in the lidar ratio (e.g., line 389), so why not partition the data in that way? That would strengthen your hypothesis. One could even repeat the boxplots that I describe above for different seasons to see if the boxes actually do move up in the dust season and down when marine aerosols have a stronger presence.*

Answer: Analysis of seasonal LR added in Fig. 15, discussion of seasonal variability of LR added into section 6 alogside with aerosol composition.

*I am not a big fan of the Scenario 1 & Scenario 2 nomenclature, as it replaces something that has meaning (excluding and including volume depolarization ratio) with something else. At least consider labeling such as Scenario E and Scenario I, as that would be easier for the reader to track.*

Answer: Nomenclature "Scenario 1" and "Scenario 2" were introduced in the related studies by Lopatin et al., 2021 which describe both methods, authors would prefer to keep them for better cross article treaceability. Clarifications were provided in the text:
**Lines 222–224:** "Table 1 summarizes instruments configurations of measurement times used for combined MPLNET AERONET retrievals using GRASP. The details of MPLNET data preparation and combined retrievals could be found in (Lopatin et al., 2021)."

*I don't believe that I have ever seen steradians abbreviated as Sr... I've always seen sr.*

Answer: Sr replaced with sr throughout the text.

*Line 159*: This is a 2nd description of the KAUST site, similar to the paragraph on line 136. There is good info in both of these paragraphs, so they should be merged and located at the beginning of Section 2 (currently line 136).

Answer: description of KAUST site was moved as suggested and updated.
**Lines 137–146**: "The KAUST campus is situated in Thuwal on the eastern coast of the Red Sea, in the western Arabian Peninsula (22.3∘ N, 39.1∘ E). The region experiences local dust storms that arise from the surrounding inland deserts (e.g., see, Kalenderski and Stenchikov, 2016), as well as distant dust from northeastern Africa through the Tokar Gap (Parajuli et al., 2020). Consequently, there is a year-round presence of desert dust in the atmosphere over the site. KAUST is unique lidar site on the Red Sea coast, and its co-location with the AERONET station allows for a more accurate retrieval of the vertical profile of aerosols (Welton et al., 2000; Parajuli et al., 2020; Lopatin et al., 2021). Additionally, KAUST has a meteorological station that performs measurements of air temperature, humidity, wind speed, and incoming short-wave and long-wave radiative fluxes. Stations that measure various parameters of interest for dust-related research, such as dust deposition rate, vertical profile, near-surface concentration, and spectral optical depth, are particularly rare across the global dust belt. The collection of these co-located data provides unique opportunity to obtain a more comprehensive understanding of dust emissions and transport in the region.
A Micro-Pulse Lidar has been in operation at KAUST site since 2014, being a part of the Micro-Pulse Lidar Network (Welton et al., 2001, 2018)."

*Line 166*: Passive tense is ambiguous here and in several places in the upcoming paragraphs. Here, the data 'was processed' using GRASP software. WHO processed the data? Consider "we processed almost three consecutive years of data... "

Answer: passive voice eliminated as suggested:
**Line 166**: "We have processed almost three consecutive years of data starting from"

*Line 181*: diluted?... or dissolved?

Answer: "diluted" replaced with more appropriate "dissolved".

*Line 205*: copped?... do you mean capped?

Answer: a typo corrected in "**cropped**"

*Line 260*: "...allowing the lidar signal to influence the photometric observations and vice versa".Do you mean 'calculations' or 'computations' instead of 'observations'?...The lidar signal won't influence the photometric observations unless you point the MPLNET at the AERONET.

Answer: passage re-phrased "...allowing the lidar signal to influence the photometric **retrievals** and vice versa."

*Line 323*: I am pretty sure that you do not mean "...nighttime liar retrievals."

Answer: typo in "lidar" corrected

*Line 407*: I don't consider Angstrom Exponent as an 'advanced aerosol product' (at least for AERONET).

Answer: passage re-phrased to "...**derived** aerosol products, such as Angström exponent ad SSA"

*Line 412*: Authors discuss potential issues associated with signal attenuation, but shouldn't that be easy to test?... just filter our high AOD cases.

Answer: After changing the scatter plots into boxplots the discussion of outliers was left out of the scope and was ommited.

*Line 513:* *(and elsewhere): 80.60% and 69.31%... that's a lot of precision. Why not round off to the nearest percentage?*

Answer: precision decimated throughout the text.

*Line 520: 6000, not 6oo0*

Answer: Corrected to 6000.

*Figure 9: Light blue is difficult to see on white. Consider a grey background.*

Answer: Colorsceme in accordance with CVD-friendly recommendations [https://www.atmospheric-measurement-techniques.net/submission.html#figurestables] was changed to make **Figures 8 and 13** more contrast.

*Table 3:* *Would be nice to see the avg values as well.*

Answer: Lines containing average values for columnar LR and AOD were added to **Table 3**

*Fig 15: Some clarification about exactly what these mode fractions mean would be helpful. For example, should the winter nights from the three panels add to 100%? It doesn't appear that way, so I am not quite sure of what these fractions mean.*

Answer: Sum of all fractions should give 100% for each particular case, althgough this does not necessarily applies to the seasonal median values. Llarifications added to the text:
**Lines 699-701:**"It should be outlined, that generally, for each particular retrieval these fractions represent 100% of aerosol by volume, however median values may not add up to 100% for each of the seasons."

*Line 683: "It should be noted, that in autumn real part of refractive index has similar values,..." needs clarification. Similar to what? Also, Autumn has stronger absorption than in Winter and similar to Spring and Summer? Is this the fine mode or coarse mode? Either way, it is difficult to reconcile this sentence with the RRI and IRI of Fig 16.*

Answer: clarification added
**Line 725**: "real part of refractive index has similar values **with winter season**"